# Variational Continual Bayesian Meta-Learning

**Qiang Zhang**[1,2,3*†] **, Jinyuan Fang**[4†] **, Zaiqiao Meng**[5,6] **, Shangsong Liang**[4,6‡] **, Emine Yilmaz**[7‡]

[1] Hangzhou Innovation Center, Zhejiang University, China
[2] College of Computer Science and Technology, Zhejiang University, China
[3] AZFT Knowledge Engine Lab, China; [4] Sun Yat-sen University, China
[5] University of Glasgow, United Kingdom
[6] Mohamed bin Zayed University of Artificial Intelligence, United Arab Emirates
[7] University College London, United Kingdom
{qiang.zhang.cs@zju.edu.cn; fangjy6@gmail.com; zaiqiao.meng@gmail.com}
{liangshangsong@gmail.com; emine.yilmaz@ucl.ac.uk}

## Abstract

Conventional meta-learning considers a set of tasks from a stationary distribution. In contrast, this paper focuses on a more complex online setting, where tasks arrive sequentially and follow a non-stationary distribution. Accordingly, we propose a Variational Continual Bayesian Meta-Learning (VC-BML) algorithm. VC-BML maintains a Dynamic Gaussian Mixture Model for meta-parameters, with the number of component distributions determined by a Chinese Restaurant Process. Dynamic mixtures at the meta-parameter level increase the capability to adapt to diverse and dissimilar tasks due to a larger parameter space, alleviating the negative knowledge transfer problem. To infer the posteriors of model parameters, compared to the previously used point estimation method, we develop a more robust posterior approximation method – structured variational inference for the sake of avoiding forgetting knowledge. Experiments on tasks from non-stationary distributions show that VC-BML is superior in transferring knowledge among diverse tasks and alleviating catastrophic forgetting in an online setting.

## 1   Introduction

Meta-learning inductively transfers knowledge among analogous low-resource tasks, i.e., tasks with scarce labeled data such as few-shot image classification, to enhance model generalization and data efficiency [1, 2]. Conventional meta-learning assumes that these tasks follow a stationary distribution. However, this assumption may be unrealistic in an online setting, where the task distribution is normally non-stationary, i.e., subsequent tasks are disparate and heterogeneous [3, 4]. In reality, the newly arriving tasks may differ from previous ones or even contradict to each other [5], e.g., the task of online landmark prediction can encounter images from unnamed classes. This leads to two issues: (1) a single set of model parameters exhibit decreased performance on disparate tasks, called as the *negative knowledge transfer* issue [6]; and (2) for fast adaptation to those disparate tasks, a model often has to dramatically change its parameters and thus often forgets previously learnt knowledge, known as the *catastrophic forgetting* issue [7]. Due to these issues, it is inefficient to use conventional meta-learning in an online setting. This motivates us to extend the conventional meta-learning to deal with the streaming low-resource tasks that follow a non-stationary distribution, such that we can dynamically update the model to effectively transfer knowledge while avoid forgetting.

---

*The work was done while at University College London

†Equal Contributions

‡Corresponding Author

35th Conference on Neural Information Processing Systems (NeurIPS 2021).

Several attempts have been made to study meta-learning for low-resource tasks from a non-stationary distribution. One branch is online meta-learning [3, 8, 9] through the perspective of regret-minimization, where the goal is to minimize the accumulative loss of the best fixed model in hindsight. These algorithms require accumulating subsequent data for training, which may not be realistic as the size of datasets often prohibits frequent batch updating. Another branch is continual learning, which avoids revisiting previous data and aims to overcome the catastrophic forgetting issue. Relevant works fell in this branch are either based on a single set of meta-parameters or a mixture of task-specific parameters. Typically, a single meta-parameter distribution [6] has a limited parameter space, which inhibits the capability to handle the boundlessly diverse tasks and incurs the negative knowledge transfer issue, leading to suboptimal performance. A mixture model has a larger parameter space and Jerfel et al. [4] build a mixture of task-specific parameter distributions, but the meta-parameters still follow delta distribution. Also, they use point estimation to infer parameters, which is prone to suffer from the catastrophic forgetting issue in an online setting.

To fill the research gap, this paper proposes a Variational Continual Bayesian Meta-Learning (VC-BML) algorithm that tackles the issues of negative knowledge transfer and catastrophic forgetting for streaming low-resource tasks. As a fully Bayesian algorithm, VC-BML assumes meta-parameters and task-specific parameters follow their respective distributions. We set out to make theoretical and empirical contributions as follows. (1) We propose meta-parameters to follow a mixture of dynamically updated distributions, each component of which is associated with a cluster of similar tasks. By assuming meta-parameters follow Gaussian distributions, we model the whole mixture with a Dynamic Gaussian Mixture Model (DGMM). Compared to a single set of meta-parameters or a mixture of task-specific parameters, dynamic mixtures of meta-parameter distributions provide one more level of flexibility in a larger parameter space and thus increase the capability to alleviate the negative knowledge transfer issue. (2) Unlike the previous work [4] that applies point estimation during the inference, which is prone to forgetting knowledge, we approximate the posterior distributions of interest by deriving a structured variational inference method. Given that, we can sample from parameter distributions and quantify the model uncertainty. (3) Finally, extensive experiments show our VC-BML algorithm outperforms seven state-of-the-art baselines on non-stationary task distributions from four benchmark datasets. It has empirically shown that the Bayesian formulation of our algorithm can alleviate negative transfer among dissimilar tasks and prevent dramatic parameter changes to overcome the catastrophic forgetting issue.

## 2   Literature Review

**Online Meta-Learning.** Meta-learning extracts transferable knowledge from a set of meta-training datasets to efficiently tackle low-resource tasks [2], such as few-shot image classification [10] and robot control [11]. Various approaches, including model-based (or black box) [12], metric-based (or non-parametric) [13], optimization-based [14] and their Bayesian counterparts [15, 16, 17, 18], have been proposed. However, these algorithms assume that tasks are from a stationary distribution, which is unrealistic in dynamic learning scenarios. To handle sequentially arriving tasks from a non-stationary distribution, online meta-learning algorithms have been under studied. Two settings for online meta-learning are identified [5]: the online-within-online setting where tasks and examples within tasks arrive sequentially, and the online-within-batch setting where tasks arrive sequentially but examples within tasks are in batch. Most of the concurrent works belong to the latter setting. From the viewpoint of regret-minimization, a Follow-The-Meta-Leader algorithm is proposed in [3] and generalized from convex to non-convex cases in [9]. In [8], the meta-learner is disentangled as a meta-hierarchical graph consisting of multiple knowledge blocks in a deterministic way. These algorithms make assumptions on regret functions and require accumulating subsequent datasets, which puts high demand for computational memory.

**Continual Learning.** Continual learning is another paradigm for sequentially arriving tasks. It avoids revisit previous data and aims to overcome catastrophic forgetting [7]. Techniques such as elastic weight consolidation [19], variational continual learning [20], online Laplace approximation [21] and brain-inspired replay [22] have been developed. Although tasks arrive sequentially, most continual learning works primarily focus on supervised learning with large-scaled annotations, which is opposed to our low-resource tasks. Continual-meta learning [23] and meta-continual learning [24] bridge the gap between meta-learning and continual learning. The former aims for quickly recover performance on previous tasks, which is a different research goal from this paper. The latter uses

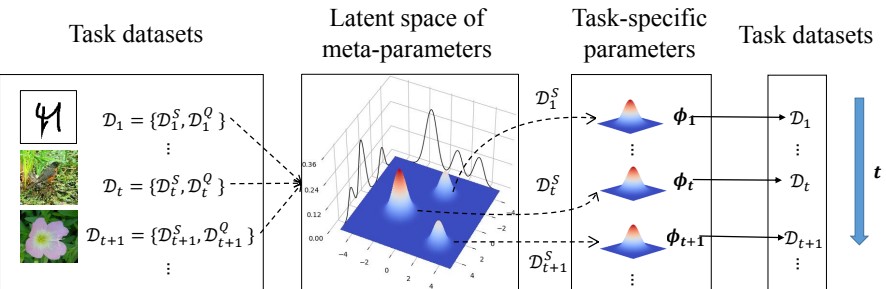

Figure 1: The framework of VC-BML. It infers (dashed line) the variational distributions of meta-parameters (Gaussian-shaped distributions) based on tasks. It then ranks candidate meta-parameter distributions in the latent space based on the posterior and select the one with highest probability. Subsequently, task-specific parameters are derived from the meta-parameters and the support set, which are efficient for making predictions (solid line).

meta-learning as a way to overcome catastrophic forgetting, but ignores the low-resource task setting. Moreover, they have been extended to online fast adaptation and knowledge accumulation [25].

Some continual learning works combine meta-learning to deal with low-resource tasks. For example, Jerfel et al. [4] propose to cluster task-specific parameter distributions and allow the meta-learner to select over a mixture of these distributions. Nonetheless, the meta-parameters still follow delta distributions and the latent variables are inferred by point estimation, which is found prone to suffer from the catastrophic forgetting issue. Yap et al. [6] make steps in Bayesian approximation to parameter distributions, but they only assumes a single meta-parameter distribution, which is not capable enough to deal with boundlessly diverse low-resource tasks in a streaming scenario. We overcome these weaknesses by representing meta-parameters with a dynamically updated mixture model and inferring latent variables via structural variational inference.

## 3 Background

### 3.1 Problem Formulation

Given sequentially arriving tasks that follow a distribution $p(\tau)$, at each time step $t$, a machine learning model receives a task $\tau_t$ with a dataset $\mathcal{D}_t$ sampled from the data distribution $p(\mathcal{D}_t|\tau_t)$. Similar to the conventional meta-learning setting [14], the dataset $\mathcal{D}_t$ is split into a support set $\mathcal{D}_t^S = \{(\boldsymbol{x}_i, y_i)\}_{i=1}^{N_t^S}$ for training and a query set $\mathcal{D}_t^Q = \{(\boldsymbol{x}_i, y_i)\}_{i=1}^{N_t^Q}$ for validation. Two fundamental elements build up our research problem. First of all, $p(\tau)$ can be non-stationary, i.e., subsequent tasks from this distribution can be disparate and heterogeneous [3, 4]. This heterogeneity leads to negative knowledge transfer when tasks are dissimilar, and often requires substantial parameter changes for fast adaption, causing the issue of catastrophic forgetting [7]. Second, $\mathcal{D}_t$ contains few shots (i.e., small $N_t^S$) that are not enough for large-scale supervised training, which motivates us to update model parameters with meta-learning. Without revisiting previous data, we aim to dynamically update the model parameters to efficiently transfer knowledge to new tasks and avoid the issue of catastrophic forgetting.

### 3.2 Preliminary

**Meta-Learning.** Meta-learning aims to discover the shared common patterns among a set of learning tasks drawn from a distribution $p(\tau)$, such that the model is able to accomplish new learning tasks with a limited number of training examples and trials. To this end, the learning procedure of meta-learning is split into two steps. The first step is to learn the meta-parameters $\boldsymbol{\theta}^\star$ from the meta-training dataset $\mathcal{D}_{\mathrm{meta-train}}$: $\boldsymbol{\theta}^\star = \arg\max_{\boldsymbol{\theta}} \log p(\mathcal{D}_{\mathrm{meta-train}} \mid \boldsymbol{\theta})$. $\boldsymbol{\theta}^\star$ can be viewed as the common knowledge among tasks from $p(\tau)$. Using $\boldsymbol{\theta}^\star$ as prior, the second step is to obtain the task-specific model parameters $\boldsymbol{\phi}^\star$ that are able to generalize well on the meta-test set $\mathcal{D}_{\mathrm{meta-test}}$ after a few trials: $\boldsymbol{\phi}^\star = \arg\max_{\boldsymbol{\phi}} \log p(\mathcal{D}_{\mathrm{meta-test}}|\boldsymbol{\phi}, \boldsymbol{\theta}^\star)$. The task-specific parameters $\boldsymbol{\phi}^\star$ can be seen as transferred knowledge from the common among a collection of tasks.

**Variational Continual Learning.** Continual learning aims to overcome the catastrophic knowledge forgetting issue when dealing with sequentially occurring tasks from a non-stationary distribution. Variational continual learning (VCL) [20] is a Bayesian approach to continual learning that offers an efficient way to infuse past knowledge via posteriors. With streaming datasets $\mathcal{D}_{1:t}$, VCL factorizes the posterior of model parameters as:

$$p\left(\boldsymbol{\theta}_t \mid \mathcal{D}_{1:t}\right) \propto p(\boldsymbol{\theta}_t) \prod_{t'=1}^{t} p\left(\mathcal{D}_{t'} \mid \boldsymbol{\theta}_t\right) \propto p\left(\mathcal{D}_t \mid \boldsymbol{\theta}_t\right) p\left(\boldsymbol{\theta}_t \mid \mathcal{D}_{1:t-1}\right). \tag{1}$$

The datasets are assumed to be independent given the model parameters $\boldsymbol{\theta}$. As we can see from Equation 1, the parameter posterior on the $(t-1)$-th dataset is consecutively taken as its prior on the $(t)$-th dataset. This motivates a recursive way of posterior approximation: starting from the first posterior $p\left(\boldsymbol{\theta}_1 \mid \mathcal{D}_1\right) \approx q(\boldsymbol{\theta}_1)$, subsequent approximations can be recursively accessed by multiplying the $(t-1)$-th posterior approximation with the $t$-th dataset likelihood, and renormalizing. More details are in Section A of Appendix.

## 4 Variational Continual Bayesian Meta-Learning

As the formulated research problem premises few shots per sequentially arriving task, VC-BML sets out to learn the meta-parameters $\boldsymbol{\theta}_t$ in an online fashion, from which the task-specific parameters $\boldsymbol{\phi}_t$ are derived. Similar to variational continual learning, the posterior distribution of $\boldsymbol{\theta}_t$ in VC-BML is given in a recursive way:

$$p\left(\boldsymbol{\theta}_t \mid \mathcal{D}_{1:t}\right) \propto p\left(\mathcal{D}_t \mid \boldsymbol{\theta}_t\right) p\left(\boldsymbol{\theta}_t \mid \mathcal{D}_{1:t-1}\right) = \left(\int p\left(\mathcal{D}_t \mid \boldsymbol{\phi}_t\right) p\left(\boldsymbol{\phi}_t \mid \boldsymbol{\theta}_t\right) d\boldsymbol{\phi}_t\right) p\left(\boldsymbol{\theta}_t \mid \mathcal{D}_{1:t-1}\right). \tag{2}$$

In the online scenario, we may encounter tasks from non-stationary task distribution, where some tasks are dissimilar or even contradicting to each other. This leads to a problem that disparate tasks require a more significant degree of adaptation. Albeit we compute a distribution of the meta-parameters, it may not be able to sufficiently adapt to a diversity of tasks. Consequently it will be inefficient if the task-specific parameters $\boldsymbol{\phi}_t$ are adapted from the simple Gaussian distribution of meta-parameters $\boldsymbol{\theta}_t$.

### 4.1 Mixture of Meta-Parameter Distributions

To improve task adaption, we propose to maintain a mixture of dynamically updated meta-parameter distributions, each of which is associated with a cluster of similar tasks. A schematic illustration is shown in Figure 1. In particular, we consider the mixture of meta-parameter distributions as a Gaussian Mixture Model (GMM). Besides, our task requires that the parameters (i.e., the means and variances) of GMM can be dynamically updated as new tasks emerges. Hence, $p(\boldsymbol{\theta}_t \mid \mathcal{D}_{1:t})$ is designed to be a DGMM as opposed to a commonly-used single static Gaussian distribution.

Let $z$ be the latent categorical variable indicating the assignment of a task to a cluster, which is equivalent to the selection of component in a mixture distribution. According to the definition of $z$, we rewrite the conditional probability:

$$p(\mathcal{D}_t \mid \boldsymbol{\theta}_t) = \int p(\mathcal{D}_t \mid \boldsymbol{\theta}_t, z_t) p(z_t) dz_t = \int \left[\int p(\mathcal{D}_t \mid \boldsymbol{\phi}_t) p(\boldsymbol{\phi}_t \mid \boldsymbol{\theta}_t^{z_t}) d\boldsymbol{\phi}_t\right] p(z_t) dz_t. \tag{3}$$

A component distribution $\boldsymbol{\theta}_t^{z_t}$ is selected from the mixture distribution of $\boldsymbol{\theta}_t$. Upon this, the distribution of task-specific parameters $\boldsymbol{\phi}_t$ can be derived. Compared to a single distribution, the mixture of meta-parameter distributions is able to increase the capacity of meta-learning to deal with more diverse tasks. However, it is non-trivial to determine the number of components in the mixture distribution to deal with the potentially boundlessly diverse tasks in the online setting.

**Chinese Restaurant Process (CRP).** To adaptively create clusters as new tasks require, we employ the Chinese Restaurant Process to flexibly determine the number of task clusters [26, 27, 28] (i.e., the number of mixture components in DGMM). This allows to create new meta-parameter distributions for disparate tasks and recall existing distributions when a new task is similar to current task clusters. Specifically, we formulate the task cluster prior as a CRP such that a new cluster is created during a

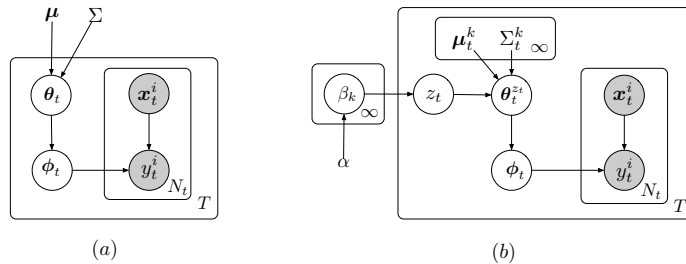

(a)                                          (b)

Figure 2: Variational Continual Bayesian Meta-Learning with (a) a single distribution of the meta-parameters; (b) a potentially infinite number of meta-parameter distributions.

trial controlled by the parameter $\alpha$. Hence the probability of a task belonging to the cluster $k$ can be given by:

$$P(z_t = k) = \begin{cases} \frac{n_k}{t-1+\alpha} & \text{if the cluster } k \text{ exists,} \\ \frac{\alpha}{t-1+\alpha} & \text{if } k \text{ is a new cluster,} \end{cases} \tag{4}$$

where $n_k$ is the expected number of tasks in the cluster $k$, and $\alpha > 0$ is a concentration parameter that controls the instantiation of a new cluster: a larger $\alpha$ will produce more clusters (and fewer tasks per cluster). Since we assume each task arrives at a timestamp, $t - 1$ indicates the total number of tasks except the current one. Hereby we use $\text{CRP}(\alpha)$ to denote Equation 4, and CRP-DGMM to denote a DGMM whose number of components is determined by CRP, which can potentially be infinite.

**Generative process of CRP-DGMM.** Mixture models need a stick-breaking representation of CRP to obtain component weights. With an infinite construction of stick-breaking representation, the probability vector $\boldsymbol{\pi} = [\pi_1, \pi_2, \cdots]$, where each element is non-negative and the elements sum to 1, is defined as:

$$\beta_k \mid \alpha \sim \text{Beta}(1, \alpha) \qquad \text{for } k = 1, \ldots, \infty, \tag{5}$$

$$\pi_k = \beta_k \prod_{l=1}^{k-1} (1 - \beta_l) \qquad \text{for } k = 1, \ldots, \infty, \tag{6}$$

which is equivalent to the weights implied by the $\text{CRP}(\alpha)$. Based on this, the generative process of CRP-DGMM is as follows:

$$z_t \mid \boldsymbol{\pi} \sim \text{Categorical}_\infty(\boldsymbol{\pi}), \tag{7}$$

$$\boldsymbol{\theta}_t^{z_t=k} \sim \mathcal{N}\left(\boldsymbol{\mu}_t^k, \Sigma_t^k\right), \tag{8}$$

$$\boldsymbol{\phi}_t \mid \boldsymbol{\theta}_t^{z_t=k} \sim p(\boldsymbol{\phi}_t \mid \boldsymbol{\theta}_t^{z_t=k}), \tag{9}$$

$$\mathcal{D}_t \mid \boldsymbol{\phi}_t \sim p(\mathcal{D}_t \mid \boldsymbol{\phi}_t), \tag{10}$$

where $\boldsymbol{\mu}_t^k$ is the mean vector and $\Sigma_t^k$ is the semi-positive covariance matrix that parameterizes the $k$-th component distribution in DGMM. Figure 2 (a) and (b) show the generative process of VC-BML with single and mixed distributions for meta-parameters respectively. For the sake of computational efficiency, a finite stick-breaking construction of CRP is used. It simply places an upper bound on the total number of mixture components $K$ in DGMM, which, if chosen reasonably, allows us to enjoy the benefits of CRP at a cheaper computational cost. The finite version of the generative process of CRP-DGMM is available in Section B of Appendix.

### 4.2   Structured Variational Inference for CRP-DGMM

The exact posterior of all latent variables, i.e., $p(\boldsymbol{\phi}_t, \boldsymbol{\theta}_t, z_t, \boldsymbol{\beta} \mid \mathcal{D}_t)$ is intractable. Due to the large dimension of latent variables, Monte Carlo sampling (e.g., Gibbs sampling) can be computationally slow [29]; the expectation maximization used in [4] only provides point estimations of latent variables. To allow efficient uncertainty quantification, we derive a structured variational inference to approximate the exact posterior. At each timestep $t$, we use only the dataset $\mathcal{D}_t$ at hand to infer latent variables including $\boldsymbol{\theta}_t$ and $\boldsymbol{\phi}_t$. We do not revisit previous datasets $\mathcal{D}_{1:t-1}$ for such inference. The exact posterior can be approximated by using the following structured variational posterior:

$$q(\boldsymbol{\phi}_t, \boldsymbol{\theta}_t, z_t, \boldsymbol{\beta} \mid \mathcal{D}_t) = q(\boldsymbol{\phi}_t \mid \boldsymbol{\theta}_t^{z_t}, \mathcal{D}_t; \boldsymbol{\lambda}_t) q(\boldsymbol{\theta}_t^{z_t} \mid \mathcal{D}_t; \boldsymbol{\psi}_t) q(z_t; \boldsymbol{\eta}_t) \times \prod_{k=1}^{K-1} q(\boldsymbol{\beta}_k; \boldsymbol{\gamma}_k), \tag{11}$$

where $\boldsymbol{\gamma}_k$, $\boldsymbol{\eta}_t$, $\boldsymbol{\psi}_t$ and $\boldsymbol{\lambda}_t$ are parameters of four variational distributions, each of which is for $\boldsymbol{\beta}_k$, $z_t$, $\boldsymbol{\theta}_t$ and $\boldsymbol{\phi}_t$ respectively. The Evidence Lower Bound (ELBO) of observations at time $t$ can be:

$$\mathcal{L}_{\text{VC-BML}}(\boldsymbol{\lambda}_t, \boldsymbol{\psi}_t, \boldsymbol{\eta}_t, \boldsymbol{\gamma}) \triangleq \mathbb{E}_{q(z_t; \boldsymbol{\eta}_t)q(\boldsymbol{\theta}_t^{z_t} | \mathcal{D}_t; \boldsymbol{\psi}_t)q(\boldsymbol{\phi}_t | \boldsymbol{\theta}_t^{z_t}, \mathcal{D}_t; \boldsymbol{\lambda}_t)} \left[ \log p(\mathcal{D}_t \mid \boldsymbol{\phi}_t) \right]$$

$$- \mathbb{E}_{q(z_t; \boldsymbol{\eta}_t)q(\boldsymbol{\theta}_t^{z_t} | \mathcal{D}_t; \boldsymbol{\psi}_t)} \left[ \text{KL} \left[ q(\boldsymbol{\phi}_t \mid \boldsymbol{\theta}_t^{z_t}, \mathcal{D}_t; \boldsymbol{\lambda}_t) || p(\boldsymbol{\phi}_t | \boldsymbol{\theta}_t^{z_t}) \right] \right]$$

$$- \mathbb{E}_{q(z_t; \boldsymbol{\eta}_t)} \left[ \text{KL} \left[ q(\boldsymbol{\theta}_t^{z_t} \mid \mathcal{D}_t; \boldsymbol{\psi}_t) || p(\boldsymbol{\theta}_t^{z_t} \mid \mathcal{D}_{1:t-1}) \right] \right]$$

$$- \mathbb{E}_{q(\boldsymbol{\beta}; \boldsymbol{\gamma})} \left[ \text{KL} \left[ q(z_t; \boldsymbol{\eta}_t) || p(z_t | \boldsymbol{\beta}) \right] \right] - \text{KL} \left[ q(\boldsymbol{\beta}; \boldsymbol{\gamma}) || p(\boldsymbol{\beta}) \right], \tag{12}$$

where $\text{KL}[\cdot || \cdot]$ is the Kullback–Leibler divergence. The ELBO of our likelihood function is represented as $\mathcal{L}_{\text{VC-BML}}(\boldsymbol{\lambda}_t, \boldsymbol{\psi}_t, \boldsymbol{\eta}_t, \boldsymbol{\gamma})$, and the variational distributions are optimized by maximizing this ELBO. We present the updates of variational parameters here; the detailed derivations are in Section C of Appendix. Pseudo codes for VC-BML are in Section D of Appendix.

**Variational distribution of $\boldsymbol{\beta}$.** By only considering the terms related to $\boldsymbol{\beta}_k$ in the ELBO, we can analytically derive its optimal variational distribution: $q(\boldsymbol{\beta}_k; \boldsymbol{\gamma}_k) = \text{Beta}(\boldsymbol{\beta}_k; \gamma_{k,1}, \gamma_{k,2})$, which is also a Beta distribution with parameters: $\gamma_{k,1} = 1 + \boldsymbol{\eta}_{t,k}$ and $\gamma_{k,2} = \alpha + \sum_{r=k+1}^{K} \boldsymbol{\eta}_{t,r}$.

**Variational distribution of $z_t$.** Similarly, the variational distribution of variable $z_t$ can be derived as: $q(z_t; \boldsymbol{\eta}_t) = \text{Categorical}_K(z_t; \boldsymbol{\eta}_t)$, which is a Categorical distribution with parameters

$$\log \boldsymbol{\eta}_{t,k} \propto \mathbb{E}_{q(\boldsymbol{\beta}; \boldsymbol{\gamma})} \left[ \boldsymbol{\pi}_k \right] - \text{KL} \left[ q(\boldsymbol{\theta}_t^{z_t=k} \mid \mathcal{D}_t; \boldsymbol{\psi}_t) || p(\boldsymbol{\theta}_t^{z_t=k} \mid \mathcal{D}_{1:t-1}) \right]$$

$$- \mathbb{E}_{q(\boldsymbol{\theta}_t^{z_t=k} | \mathcal{D}_t; \boldsymbol{\psi}_t)} \left[ \text{KL} \left[ q(\boldsymbol{\phi}_t \mid \boldsymbol{\theta}_t^{z_t=k}, \mathcal{D}_t; \boldsymbol{\lambda}_t) || p(\boldsymbol{\phi}_t | \boldsymbol{\theta}_t^{z_t=k}) \right] \right], \tag{13}$$

where s.t. $\sum_{k=1}^{K} \boldsymbol{\eta}_{t,k} = 1$. $q(\boldsymbol{\beta}; \boldsymbol{\gamma}) = \prod_{k=1}^{K-1} q(\boldsymbol{\beta}_k; \boldsymbol{\gamma}_k)$ is the variational distribution of $\boldsymbol{\beta}$.

**Variational distribution of $\boldsymbol{\theta}_t$.** The meta-parameter distribution is assumed to contain a mixture of component distributions. We assume the variational distribution of the $k$-th component to be:

$$q\left( \boldsymbol{\theta}_t^{z_t=k} \mid \mathcal{D}_t; \boldsymbol{\psi}_t \right) = \mathcal{N}\left( \boldsymbol{m}_t^k, \Lambda_t^k \right), \tag{14}$$

which is a Gaussian distribution with parameters $\boldsymbol{\psi}_t^{z_t=k} = \{\boldsymbol{m}_t^k, \Lambda_t^k\}$. $\boldsymbol{m}_t^k$ is the mean vector and $\Lambda_t^k$ is the semi-positive covariance matrix of the $k$-th component at the timestamp $t$. Without loss of generality, we assume the prior of $\boldsymbol{\theta}_t^k$ is given by the variational posterior of $\boldsymbol{\theta}_{t-1}^k$, i.e., $\boldsymbol{\mu}_t^k = \boldsymbol{m}_{t-1}^k, \Sigma_t^k = \Lambda_{t-1}^k$. The variational distribution of $\boldsymbol{\theta}_t^k$ can be recursively updated by maximizing the ELBO with stochastic back propagation [30].

**Variational distribution of $\boldsymbol{\phi}_t$.** For $q(\boldsymbol{\phi}_t \mid \boldsymbol{\theta}_t^{z_t}, \mathcal{D}_t; \boldsymbol{\lambda}_t)$, $\boldsymbol{\phi}_t$ denotes the weights of a deep neural network and $\boldsymbol{\lambda}_t$ denotes the variational parameters (i.e., means and standard deviations). Due to the high dimension of $\boldsymbol{\phi}_t$, it is computationally difficult to learn variational parameters $\boldsymbol{\lambda}_t$. Hence, we resort to amortized variational inference (AVI) [17], where a global initialization $\boldsymbol{\theta}_t^{z_t}$ is used to produce $\boldsymbol{\lambda}_t$ from $\mathcal{D}_t^S$ ($\mathcal{D}_t$ is split to a support set $\mathcal{D}_t^S$ for adapting task-specific parameters $\boldsymbol{\phi}_t$ and a query set $\mathcal{D}_t^Q$ for task evaluation) by conducting several steps of gradient descents:

$$q(\boldsymbol{\phi}_t \mid \boldsymbol{\theta}_t^{z_t}, \mathcal{D}_t; \boldsymbol{\lambda}_t) = q(\boldsymbol{\phi}_t; SGD_J(\boldsymbol{\theta}_t^{z_t}, \mathcal{D}_t^S, \epsilon)), \tag{15}$$

where $J$ is the gradient descent steps and $\epsilon$ is the learning rate. The procedure details of stochastic gradient descent, i.e., $SGD_J(\boldsymbol{\theta}_t^{z_t}, \mathcal{D}_t^S, \epsilon)$, are presented in Section C of Appendix.

**Down-weighting KL-divergence.** Maximizing the ELBO will recover true posteriors if the approximating family is sufficiently rich. However, the simple families used in practice typically lead to poor test-set performance. Some "Annealing" techniques, such as the Cyclical Annealing [31] and data augmented down-weighting [32, 33], can alleviate this issue by introducing a factor $\nu$ on the KL-divergence term. In this paper, following [32, 33], we introduce an additional hyperparameter $\nu(0 < \nu \leq 1)$ to down-weight the KL-divergence term of $\boldsymbol{\theta}_t$.

Table 1: Mean meta-test accuracies (%) at each meta-training stage.

| Algorithms | Omniglot | CIFAR-FS | *mini*Imagenet | VGG-Flowers |
|---|---|---|---|---|
| TOE | $98.43 \pm 0.48$ | $80.55 \pm 1.39$ | $67.55 \pm 1.15$ | $64.57 \pm 1.71$ |
| TFS | $98.22 \pm 0.50$ | $82.39 \pm 1.30$ | $71.63 \pm 1.60$ | $65.50 \pm 1.73$ |
| FTML | $99.11 \pm 0.16$ | $82.33 \pm 1.31$ | $71.97 \pm 1.41$ | $65.46 \pm 1.63$ |
| OSML | $99.30 \pm 0.29$ | $83.91 \pm 1.19$ | $75.13 \pm 1.36$ | $70.89 \pm 1.44$ |
| DPMM | $\mathbf{99.33 \pm 0.28}$ | $83.52 \pm 1.21$ | $73.96 \pm 1.37$ | $70.09 \pm 1.40$ |
| OSAKA | $99.01 \pm 0.29$ | $82.87 \pm 1.24$ | $74.34 \pm 1.48$ | $67.18 \pm 1.64$ |
| BOMVI | $96.25 \pm 0.66$ | $83.05 \pm 1.25$ | $74.98 \pm 1.43$ | $76.56 \pm 1.46$ |
| VC-BML | $98.62 \pm 0.38$ | $\mathbf{86.95 \pm 1.15}$ | $\mathbf{78.29 \pm 1.34}$ | $\mathbf{79.37 \pm 1.43}$ |

## 4.3 Discussion

There is a recent work [4] that uses a mixture model in meta-learning. However, our proposed VC-BML differs from this work in three aspects that consequently lead to performance improvement. First, the meta-parameters in our model are represented by a mixture of Gaussians compared to delta distributions in [4]. Compared to delta-distributions, Gaussians generally enable a larger capacity in tackling related but moderately different tasks in one cluster, and can capture some uncertainty that delta-distributions with zero variance cannot do. Second, we derive a structural variational inference method to approximate the posterior while Jerfel et al. [4] only provide point estimates based on maximum a posteriori (MAP). In a streaming setting, point estimates on the past cannot imply doing well on future data as underlying generation functions can be ambiguous even given some prior information. Third, despite in principle the posterior can be regularized by the prior in [4], it neither explicitly explains how to achieve this goal nor adopts this regularization method in their model (see Algorithm 1 in [4]). In contrast, the KL-divergence in the VI framework in our model can be naturally considered as the regularization term. By tuning the weight of this regularization term, we can reduce overfitting to the incoming data, leading to improved performance (and we have proved this in the Experiments section).

## 5 Experiments

We verify the effectiveness of VC-BML via experimental comparison and analysis. In particular, we answer three research questions: (**RQ1**) Can VC-BML outperform baselines on heterogeneous datasets with non-stationary task distributions? (**RQ2**) How does the number of mixture components $K$ affect results? (**RQ3**) What is the contribution of each component in our VC-BML?

To evaluate VC-BML on non-stationary task distributions, seven state-of-the-art meta-learning and continual learning algorithms are chosen as baselines: (1) Train-On-Everything (TOE), a naïve baseline that, at each time step $t$, the meta-parameters are re-initialized and meta-trained on all available data $\mathcal{D}_{1:t}$; (2) Train-From-Scratch (TFS), another intuitive approach that, at each time step $t$, the meta-parameters are re-initialized and meta-trained on the current data $\mathcal{D}_t$; (3) FTML [3], which utilizes the Follow the Leader algorithm [34] to minimize the regret of meta-learner; (4) DPMM [4], which uses a Dirichlet process mixture to model the latent task distribution; (5) OSML [8], which constructs meta-knowledge pathways between knowledge blocks to quickly adapt to new tasks; (6) OSAKA [25], which aims at online adaption to new tasks and fast remembering; (7) BOMVI [6], which uses a Bayesian approach to meta- and continual learning with variational inference.

Similar to previous works [4, 25], we conduct experiments on four datasets: Omniglot [35], CIFAR-FS [36], *mini*Imagenet [37] and VGG-Flowers [38]. Different datasets correspond to disparate tasks, and hence tasks chronologically sampled from these four datasets follow a non-stationary distribution. Consistently, we consider the conventional 5-way 5-shot image classification task, treat 5 classes randomly sampled from a dataset as a task and generate a sequence of tasks by sampling the classes. We *sequentially* meta-train models on tasks sampled from the meta-training sets of these four datasets. That is, we firstly meta-train models on tasks sampled from the meta-training set of Omniglot, then proceed to the next dataset. Performance is evaluated on its meta-test set. We tune the hyperparameters based on the validation sets. More details can be found in Section E of Appendix.

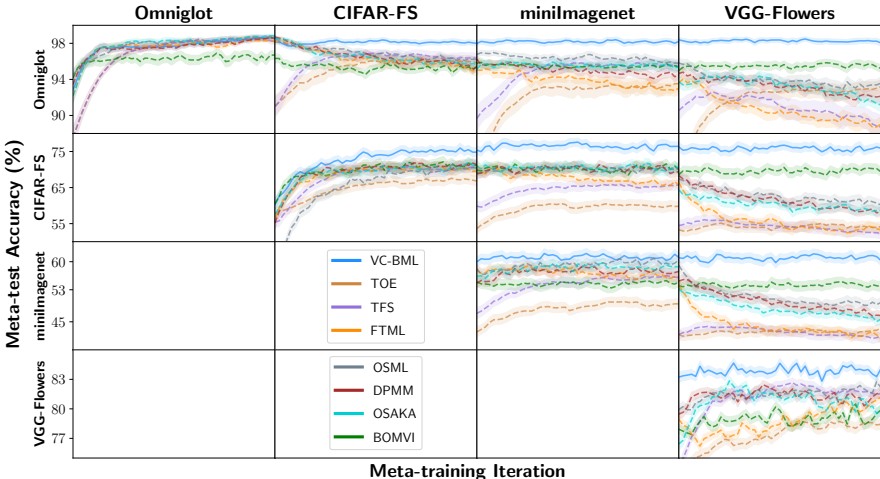

Figure 3: The evolution of meta-test accuracies (%) when algorithms are sequentially meta-trained on different datasets. Each row represents the meta-test accuracies for a dataset over cumulative number of training iterations.

## 5.1 RQ1: Performance on Non-Stationary Task Distribution

We report the mean meta-test accuracies on previously learned datasets (i.e., $\mathcal{D}_{1:t}$) at each meta-training stage in Table 1. VC-BML achieves the best performance when sequentially meta-trained on different datasets (CIFAR-FS, *mini*Imagenet and VGG-Flowers), which indicates that VC-BML can better handle tasks from a non-stationary distribution. We suspect the reason why VC-BML does not obtain the best performance on Omniglot is the over regularization caused by the KL-divergence in the ELBO. We verify this hypothesis in Section F.4 of Appendix. To obtain a more vivid presentation of knowledge transfer, we visualize the evolution of meta-test accuracies at all timestamps in Figure 3. Although FTML, OSML, DPMM and OSAKA achieve good performance on tasks from a stationary distribution (i.e., the same dataset), their performance degrades when the task distribution shifts to new datasets. In contrast, the meta-test accuracies of VC-BML on different datasets at different meta-training stages are stable and consistent. This shows VC-BML alleviates the problem of negative knowledge transfer.

To obtain deeper insights of catastrophic forgetting, we report in Table 2 the meta-test accuracies on each dataset at the beginning (Omniglot) and ending (VGG-Flowers) of meta-training stage. The full results at each meta-training stage can be found in Section F.1 of Appendix. It's shown that parametric algorithms (FTML, OSML, DPMM and OSAKA) suffer from a severer forgetting problem than Bayesian ones (BOMVI and VC-BML). For example, on Omniglot dataset, the accuracies of parametric algorithms decrease from 99.01%-99.33% to 87.84%-93.14%, while the accuracies of Bayesian algorithms barely change (from 96.25%-98.62% to 96.60%-97.47%). Similar results can be found on CIFAR-FS and *mini*Imagenet datasets (see Section F.1 of Appendix for details). This can be explained that the parametric algorithms allow to dramatically change parameters to quickly adapt to new tasks and thus forget previous knowledge, while the KL term in Bayesian algorithms can act as a regularizer to prevent such forgetting. Moreover, when only meta-trained on Omniglot, despite the performance of Bayesian algorithms on Omniglot is not the best, they always outperforms parametric counterparts on unseen datasets (i.e., CIFAR-FS, *mini*Imagenet and VGG-Flowers). This is because parametric algorithms are tailored for specific task distributions while Bayesian algorithms allow for stronger generalization on tasks from unknown distributions. Therefore, Bayesian algorithms are reckoned to be more suitable for the online scenario with non-stationary task distributions. The reason why our VC-BML outperforms BOMVI is that BOMVI suffers from negative knowledge transfer when handling different task distributions, as it only maintains a single meta-parameter distribution.

To show the findings are irrelevant to the meta-training orders, we conduct further experiments by meta-training algorithms with different dataset orders. Results are shown in Section F.2 of Appendix.

## 5.2 RQ2: Number of Mixture Components

For the sake of computational efficiency, we use a finite stick-breaking construction of CRP, where we place an upper bound on the number of mixture components, i.e., K. To study the impact of $K$, we conduct experiments by varying $K$ and report their mean meta-test accuracies on previously learned

Table 2: Meta-test accuracies (%) on each dataset at the beginning (Omniglot) and ending (VGG-Flowers) of meta-training stages. "Average" represents the mean meta-test accuracies on four datasets. (The best performance per dataset per meta-training stage is marked with boldface and the second best is marked with underline.)

| Meta-training Stage | Algorithms | Omniglot | CIFAR-FS | *mini*Imagenet | VGG-Flowers | Average |
|---|---|---|---|---|---|---|
| Omniglot | TOE | $98.43 \pm 0.48$ | $42.83 \pm 1.77$ | $33.09 \pm 1.44$ | $64.19 \pm 2.11$ | $59.64 \pm 1.45$ |
| | TFS | $98.22 \pm 0.50$ | $43.36 \pm 1.91$ | $33.49 \pm 1.38$ | $63.18 \pm 2.05$ | $59.56 \pm 1.46$ |
| | FTML | $99.11 \pm 0.16$ | $42.48 \pm 1.88$ | $34.69 \pm 1.22$ | $57.32 \pm 2.06$ | $58.40 \pm 1.33$ |
| | OSML | $\underline{99.30 \pm 0.29}$ | $41.20 \pm 1.82$ | $32.52 \pm 1.33$ | $55.39 \pm 2.19$ | $57.10 \pm 1.41$ |
| | DPMM | $\mathbf{99.33 \pm 0.28}$ | $42.31 \pm 1.62$ | $34.40 \pm 1.61$ | $57.58 \pm 2.16$ | $58.45 \pm 1.42$ |
| | OSAKA | $99.01 \pm 0.29$ | $43.24 \pm 1.79$ | $35.44 \pm 1.35$ | $60.13 \pm 2.08$ | $59.46 \pm 1.38$ |
| | BOMVI | $96.25 \pm 0.66$ | $\underline{48.22 \pm 1.94}$ | $\underline{37.83 \pm 1.45}$ | $66.24 \pm 1.98$ | $\underline{63.06 \pm 1.51}$ |
| | VC-BML | $98.62 \pm 0.38$ | $\mathbf{50.86 \pm 1.94}$ | $\mathbf{38.11 \pm 1.51}$ | $\mathbf{68.63 \pm 2.08}$ | $\mathbf{64.06 \pm 1.48}$ |
| VGG-Flowers | TOE | $93.12 \pm 1.18$ | $50.07 \pm 2.20$ | $41.97 \pm 1.74$ | $73.13 \pm 1.71$ | $64.57 \pm 1.71$ |
| | TFS | $87.84 \pm 1.89$ | $51.99 \pm 1.89$ | $41.53 \pm 1.74$ | $80.62 \pm 1.41$ | $65.50 \pm 1.73$ |
| | FTML | $87.01 \pm 1.35$ | $52.29 \pm 1.90$ | $41.43 \pm 1.67$ | $81.10 \pm 1.58$ | $65.46 \pm 1.63$ |
| | OSML | $\underline{93.14 \pm 0.90}$ | $59.74 \pm 1.85$ | $48.44 \pm 1.60$ | $\underline{82.25 \pm 1.43}$ | $70.89 \pm 1.44$ |
| | DPMM | $92.71 \pm 0.76$ | $58.76 \pm 1.83$ | $46.95 \pm 1.54$ | $81.92 \pm 1.48$ | $70.09 \pm 1.40$ |
| | OSAKA | $90.05 \pm 1.13$ | $53.37 \pm 2.01$ | $44.93 \pm 1.68$ | $80.37 \pm 1.73$ | $67.18 \pm 1.64$ |
| | BOMVI | $96.60 \pm 0.55$ | $\underline{70.16 \pm 1.84}$ | $\underline{57.94 \pm 1.65}$ | $81.54 \pm 1.81$ | $\underline{76.56 \pm 1.46}$ |
| | VC-BML | $\mathbf{97.47 \pm 0.73}$ | $\mathbf{75.20 \pm 1.80}$ | $\mathbf{61.43 \pm 1.49}$ | $\mathbf{83.39 \pm 1.73}$ | $\mathbf{79.37 \pm 1.44}$ |

Table 3: Mean meta-test accuracies (%) for VC-BML with different $K$ at each meta-training stage.

| | Omniglot | CIFAR-FS | *mini*Imagenet | VGG-Flowers |
|---|---|---|---|---|
| K=1 | $98.32 \pm 0.57$ | $81.04 \pm 1.48$ | $71.76 \pm 1.67$ | $73.09 \pm 1.54$ |
| K=2 | $98.59 \pm 0.49$ | $85.28 \pm 1.15$ | $76.48 \pm 1.28$ | $78.23 \pm 1.42$ |
| K=4 | $98.35 \pm 0.52$ | $85.91 \pm 1.15$ | $78.07 \pm 1.40$ | $79.08 \pm 1.52$ |
| K=6 | $98.62 \pm 0.38$ | $86.24 \pm 1.09$ | $78.29 \pm 1.01$ | $79.37 \pm 1.44$ |

datasets, i.e., $\mathcal{D}_{1:t}$, at different meta-training stages. The results in Table 3 show that VC-BML has the worst performance when $K = 1$, which suggests that maintaining a single meta-parameter distribution is insufficient to diverse tasks. The performance of VC-BML improves as $K$ gets larger, which indicates the benefit of instantiating more meta-parameter distributions. Note that the improvement of VC-BML is insignificant when $K > 4$ (the actual number of task distributions). More results are available in Section F.3 of Appendix.

To verify that VC-BML is able to identify different task distributions, we visualize the posterior of $z$ (the latent task cluster assignment) after meta-training VC-BML with $K = 6$ on the four datasets. Specifically, we randomly sample 100 tasks from the meta-test set of each dataset and calculate the posteriors, which is presented in Figure 4. The results show that VC-BML can recognize different task distributions with separate mixture components. For example, component 1 corresponds to Omniglot while the component 4 corresponds to VGG-Flowers. This can help explain the superiority of VC-BML when modeling non-stationary task distributions, as it can store knowledge about different task distributions in different mixture components, and recall these knowledge when similar tasks emerge. Moreover, the low posteriors of component 5 and 6 suggest that VC-BML only instantiates new meta-parameter distributions when disparate tasks are discovered. After training DPMM on the four experimental datasets, there are actually 4 task clusters, which is the same as the effective number of clusters in our model (although we set the upper bound of cluster number to be 6).

## 5.3 RQ3: Ablation Study

We conduct ablation experiments to analyze the reasons for the performance improvement, which have been discussed in Section 4.3. To examine the effect of each component, we introduce three variants of our VC-BML: VC-BML-Delta, where the variance of GMM is fixed to be a low value ($10^{-5}$) and hence GMM is converted to a mixture of delta distribution; VC-BML-MAP, where the structured VI for inference is replaced with MAP; and VC-BML-NOR, where the weight of KL-divergence is not tuned and set to be 1. The experimental results are shown in Table 4, where the evaluation results are obtained in the same way as those in Table 1. We first examine the contribution of GMM over a mixture of delta distributions by comparing VC-BML-Delta and VC-BML. We find that the performance of DPMM and VC-BML-Delta is quite similar, but lower than that of VC-BML on three datasets (CIFAR-FS, *mini*Imagenet and VGG-Flowers). This comparison quantifies the contribution of GMM over a mixture of delta distributions. We then examine the advantage of structured VI

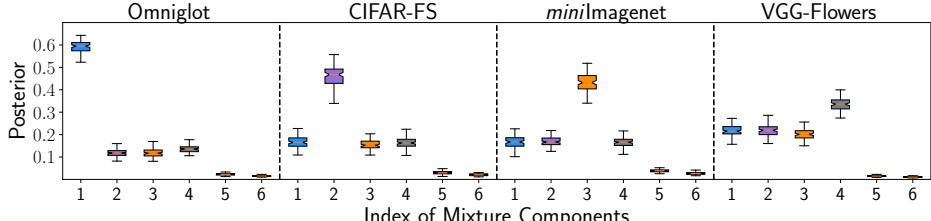

Figure 4: Each panel (column) represents the posterior of different mixture components on a dataset. A higher posterior represents a higher probability of assigning a task to a specific task cluster.

Table 4: Model ablation study on all evaluation datasets.

| Model | Omniglot | CIFAR-FS | *mini*Imagenet | VGG-Flowers |
|---|---|---|---|---|
| DPMM | $99.33 \pm 0.28$ | $83.52 \pm 1.21$ | $73.96 \pm 1.37$ | $70.09 \pm 1.40$ |
| VC-BML-Delta | $99.48 \pm 0.16$ | $84.43 \pm 1.05$ | $75.45 \pm 1.33$ | $71.27 \pm 1.52$ |
| VC-BML-MAP | $99.50 \pm 0.16$ | $84.98 \pm 1.11$ | $77.46 \pm 1.38$ | $76.27 \pm 1.51$ |
| VC-BML-NOR | $94.81 \pm 0.51$ | $80.76 \pm 0.99$ | $70.38 \pm 1.46$ | $66.88 \pm 1.32$ |
| VC-BML | $98.62 \pm 0.38$ | $86.95 \pm 1.15$ | $78.29 \pm 1.34$ | $79.37 \pm 1.43$ |

over MAP by comparing VC-BML-MAP and VC-BML. From the table, we find that VC-BML outperforms VC-BML-MAP on three out of four datasets. The performance difference shows the quantified contribution of VI in our model. In addition, our model is worse than VC-BML-Delta and VC-BML-MAP on the Omniglot dataset. This is, perhaps, due to the influence of KL-divergence and its weight in the VC-BML. These comparisons motivate us to properly weigh the KL-divergence in the VC-BML model, which consequently produced better performance on Omniglot as we showed in Section F.4 of the Appendix. Besides, in Table 2 of the Appendix, we show the optimal weights of KL-divergence that leads to the best performance on all datasets. To further examine the effects of weighting KL-divergence, we compare the performance of VC-BML-NOR and VC-BML. We notice that VC-BML-NOR performs much worse than VC-BML. Thus, if the KL-divergence is not properly scaled, the model is actually learning nothing but forcing the posterior to collapse to the prior, leading to the severe mode collapse problem.

## 6 Limitations

This paper maintains a mixture of Gaussian distributions for meta-parameters to deal with diverse tasks in a streaming scenario. One limitation comes from the added space complexity. As conventional algorithms maintain one replica of meta-parameters (either themselves or their distribution parameters such as means and variances) for all tasks, we need to allocate more space to store $K - 1$ meta-parameter replicas for those disparate tasks.

## 7 Conclusion

This paper deals with low-resource streaming tasks that follow non-stationary distributions. To avoid knowledge forgetting and negative transfer, we propose VC-BML, a Variational Continual Bayesian Meta-Learning algorithm, where the meta-parameters follow a mixture of dynamic Gaussians as opposed to the commonly-used single static Gaussian. In particular, we employ the Chinese Restaurant Process to adaptively determine the number of mixture components. To approximate the intractable posterior of interest, we develop a structural variational inference method. Extensive experiments show that our algorithm outperforms state-of-the-art baselines when adapting to diverse tasks and alleviating knowledge forgetting in an online setting with non-stationary task distributions.

As VC-BML can be widely applied to various tasks, it could exert potential negative social impacts. Although, to the best of our knowledge, we do not see a direct path to negative applications, if employed for malicious purposes, it can facilitate the development of socially-threatening models.

## Acknowledgments

This research was partially supported by the National Natural Science Foundation of China (Grant No. 61906219) and the EPSRC Fellowship entitled "Task Based Information Retrieval," grant reference number EP/P024289/1. We acknowledge the support of NVIDIA Corporation with the donation of the Titan Xp GPU.

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
