# Appendix: Variational Continual Bayesian Meta-Learning

**Qiang Zhang**[1,2,3*†], **Jinyuan Fang**[4†], **Zaiqiao Meng**[5,6], **Shangsong Liang**[4,6‡], **Emine Yilmaz**[7‡]
[1] Hangzhou Innovation Center, Zhejiang University, China
[2] College of Computer Science and Technology, Zhejiang University, China
[3] AZFT Knowledge Engine Lab, China; [4] Sun Yat-sen University, China
[5] University of Glasgow, United Kingdom
[6] Mohamed bin Zayed University of Artificial Intelligence, United Arab Emirates
[7] University College London, United Kingdom
{qiang.zhang.cs@zju.edu.cn; fangjy6@gmail.com; zaiqiao.meng@gmail.com}
{liangshangsong@gmail.com; emine.yilmaz@ucl.ac.uk}

## A   Variational Continual Learning

In variational continual learning, the posterior distribution of interest is frequently intractable and approximation is required. A projection operation is usually required to take the intractable unnormalized distribution $p^*(\boldsymbol{\theta})$ and returns a tractable normalized approximation $q(\boldsymbol{\theta})$, that is $q(\boldsymbol{\theta}) = \text{proj}\,(p^*(\boldsymbol{\theta}))$. With the recursive updating of the posterior, we have

$$p\left(\boldsymbol{\theta}_t \mid \mathcal{D}_{1:t}\right) \approx q(\boldsymbol{\theta}_t) = \text{proj}\,(q(\boldsymbol{\theta}_{t-1})p\left(\mathcal{D}_t \mid \boldsymbol{\theta}_t\right)). \tag{16}$$

The field of approximate inference provides several choices for the projection operation including i) Laplace's approximation, ii) variational KL minimization, iii) moment matching, and iv) importance sampling. Let $q(\boldsymbol{\theta}_t)$ be parameterized by $\boldsymbol{\psi}_t$, that is, $q(\boldsymbol{\theta}_t) = q(\boldsymbol{\theta}; \boldsymbol{\psi}_t)$. The online VI approach is implemented by minimizing a variational KL divergence,

$$
\begin{aligned}
&\text{KL}\left[q(\boldsymbol{\theta}_t) \| q(\boldsymbol{\theta}_{t-1})p\left(\mathcal{D}_t \mid \boldsymbol{\theta}_t\right)\right] \\
&= \int_{-\infty}^{+\infty} q(\boldsymbol{\theta}_t) \log \frac{q(\boldsymbol{\theta}_t)}{q(\boldsymbol{\theta}_{t-1})p\left(\mathcal{D}_t \mid \boldsymbol{\theta}_t\right)} d\boldsymbol{\theta}_t \\
&= \text{KL}\left[q(\boldsymbol{\theta}_t) \| q(\boldsymbol{\theta}_{t-1})\right] - \mathbb{E}_{q(\boldsymbol{\theta}_t)}\left[\log p\left(\mathcal{D}_t \mid \boldsymbol{\theta}_t\right)\right] \\
&= \mathcal{L}_{\text{VCL}}(\boldsymbol{\psi}_t).
\end{aligned}
\tag{17}
$$

This is equivalent to maximizing the likelihood $\log p(\mathcal{D}_t \mid \mathcal{D}_{1:t-1})$.

$$
\begin{aligned}
&\log p(\mathcal{D}_t \mid \mathcal{D}_{1:t-1}) \\
&= \log \int p(\mathcal{D}_t \mid \boldsymbol{\theta}_t)p(\boldsymbol{\theta}_t \mid \mathcal{D}_{1:t-1})d\boldsymbol{\theta}_t \\
&\geq \mathbb{E}_{q(\boldsymbol{\theta}_t)}\left[\log p\left(\mathcal{D}_t \mid \boldsymbol{\theta}_t\right)\right] - \text{KL}\left[q(\boldsymbol{\theta}_t) \| p(\boldsymbol{\theta}_t \mid \mathcal{D}_{1:t-1}))\right] \\
&= \mathbb{E}_{q(\boldsymbol{\theta}_t)}\left[\log p\left(\mathcal{D}_t \mid \boldsymbol{\theta}_t\right)\right] - \text{KL}\left[q(\boldsymbol{\theta}_t) \| q(\boldsymbol{\theta}_{t-1})\right] \\
&= -\text{KL}\left[q(\boldsymbol{\theta}_t) \| q(\boldsymbol{\theta}_{t-1})p\left(\mathcal{D}_t \mid \boldsymbol{\theta}_{t-1}\right)\right],
\end{aligned}
\tag{18}
$$

where $q(\boldsymbol{\theta}_{t-1}) = p(\boldsymbol{\theta}_t \mid \mathcal{D}_{1:t-1})$ means the prior of $\boldsymbol{\theta}_t$ equals the variational posterior of $\boldsymbol{\theta}_{t-1}$.

---

*The work was done while at University College London

†Equal Contributions

‡Corresponding Author

35th Conference on Neural Information Processing Systems (NeurIPS 2021).

Therefore the optimum $q(\boldsymbol{\theta}_t)$ is

$$q(\boldsymbol{\theta}_t) = \arg\min_{q \in \mathcal{Q}} \mathrm{KL}\left[q(\boldsymbol{\theta}_t) \| \frac{1}{Z_t} q(\boldsymbol{\theta}_{t-1}) p\left(\mathcal{D}_t \mid \boldsymbol{\theta}_t\right)\right],$$

$$\text{for } t = 1, 2, \ldots, T. \tag{19}$$

The zeroth approximate distribution is defined to be the prior, $q(\boldsymbol{\theta}_0) = p(\boldsymbol{\theta})$. $Z_t$ is the intractable normalizing constant and is not required to compute the optimum.

## B   Finite Stick-Breaking Construction of CRP

A truncated (or finite) version of the stick-breaking construction is used to reap the benefits of Chinese Restaurant Process at a cheaper computational cost. The finite stick-breaking model simply places an upper bound on the number of mixture components and bypasses the need for varying number of mixture components. Such a construction of the CRP can be given as follows:

$$\beta_k \mid \alpha \sim \mathrm{Beta}(1, \alpha) \qquad \text{for } k = 1, \ldots, K-1,$$

$$\pi_k = \beta_k \prod_{l=1}^{k-1} (1 - \beta_l) \quad \text{for } k = 1, \ldots, K-1, \tag{20}$$

$$\pi_K = \prod_{l=1}^{K-1} (1 - \beta_l).$$

Here on, for brevity, $\boldsymbol{\pi} = \mathrm{stickbreak}(\boldsymbol{\beta})$ will denote the above transformations, where we define vectors $\boldsymbol{\beta} = [\beta_k]_{1 \leq k \leq K-1}$ and $\boldsymbol{\pi} = [\pi_k]_{1 \leq k \leq K}$, since $\pi$ is the prior parameters of the $K$-dimensional category variable $z_t$.

With the help of the finite stick-breaking construction of the CRP, we show the generative process of CRP-DGMM for Variational Continual Bayesian Meta-Learning as follows:

$$z_t \mid \boldsymbol{\pi} \sim \mathrm{Categorical}_K(\boldsymbol{\pi}),$$

$$\boldsymbol{\theta}_t^{z_t=k} \sim \mathcal{N}\left(\boldsymbol{\mu}_t^k, \Sigma_t^k\right),$$

$$\boldsymbol{\phi}_t \mid \boldsymbol{\theta}_t^{z_t=k} \sim p(\boldsymbol{\phi}_t \mid \boldsymbol{\theta}_t^{z_t=k}), \tag{21}$$

$$\mathcal{D}_t \mid \boldsymbol{\phi}_t \sim p(\mathcal{D}_t \mid \boldsymbol{\phi}_t),$$

where $\boldsymbol{\mu}_t^k$ and $\Sigma_t^k$ are the the mean vector and the semi-positive covariance matrix that parameterize the $k$-th component distribution in DGMM.

## C   Derivation for Variational Posterior

The posterior after seeing the $t$-th dataset is $p\left(\boldsymbol{\theta}_t \mid \mathcal{D}_{1:t}\right) \propto p\left(\mathcal{D}_t \mid \boldsymbol{\theta}_t\right) p\left(\boldsymbol{\theta}_t \mid \mathcal{D}_{1:t-1}\right)$ where

$$p(\mathcal{D}_t \mid \boldsymbol{\theta}_t) = \int p(\mathcal{D}_t \mid \boldsymbol{\theta}_t, z_t) p(z_t) dz_t$$

$$= \int \left[\int p(\mathcal{D}_t \mid \boldsymbol{\phi}_t) p(\boldsymbol{\phi}_t \mid \boldsymbol{\theta}_t^{z_t}) d\boldsymbol{\phi}_t\right] p(z_t) dz_t$$

$$= \int \int \left[\int p(\mathcal{D}_t \mid \boldsymbol{\phi}_t) p(\boldsymbol{\phi}_t \mid \boldsymbol{\theta}_t^{z_t}) d\boldsymbol{\phi}_t\right] p(z_t \mid \boldsymbol{\beta}) p(\boldsymbol{\beta}) dz_t d\boldsymbol{\beta}. \tag{22}$$

Similar to Equation 18 in the continual learning setting, what we can do is to use handy $\mathcal{D}_t$ to infer latent variables. Since the exact posteriors of interest are intractable, we resort to approximate variational inference and define the variational distributions of these latent variables as:

$$q(\boldsymbol{\phi}_t, \boldsymbol{\theta}_t, z_t, \boldsymbol{\beta} \mid \mathcal{D}_t) = q(\boldsymbol{\phi}_t \mid \boldsymbol{\theta}_t^{z_t}, \mathcal{D}_t; \boldsymbol{\lambda}_t) q(\boldsymbol{\theta}_t^{z_t} \mid \mathcal{D}_t; \boldsymbol{\psi}_t) q(z_t; \boldsymbol{\eta}_t) \times \prod_{k=1}^{K-1} q(\beta_k; \boldsymbol{\gamma}_k). \tag{23}$$

As a result, the ELBO of logarithmic marginal likelihood of observations at time $t$ can be derived as:

$$\log p(\mathcal{D}_t \mid \mathcal{D}_{1:t-1})$$

$$= \log \iint p(\mathcal{D}_t \mid z_t, \mathcal{D}_{1:t-1})p(z_t|\boldsymbol{\beta})p(\boldsymbol{\beta})dz_t d\boldsymbol{\beta}$$

$$\geq \mathbb{E}_{q(z_t;\boldsymbol{\eta}_t)}\Big[\log p(\mathcal{D}_t \mid z_t, \mathcal{D}_{1:t-1})\Big] - \mathbb{E}_{q(\boldsymbol{\beta};\boldsymbol{\gamma})}\Big[\mathrm{KL}\big[q(z_t;\boldsymbol{\eta}_t)||p(z_t|\boldsymbol{\beta})\big]\Big] - \mathrm{KL}\big[q(\boldsymbol{\beta};\boldsymbol{\gamma})||p(\boldsymbol{\beta})\big]$$

$$\geq \mathbb{E}_{q(z_t;\boldsymbol{\eta}_t)q(\boldsymbol{\theta}_t^{z_t}|\mathcal{D}_t;\boldsymbol{\psi}_t)q(\boldsymbol{\phi}_t|\boldsymbol{\theta}_t^{z_t},\mathcal{D}_t;\boldsymbol{\lambda}_t)}\Big[\log p(\mathcal{D}_t \mid \boldsymbol{\phi}_t)\Big]$$

$$- \mathbb{E}_{q(z_t;\boldsymbol{\eta}_t)q(\boldsymbol{\theta}_t^{z_t}|\mathcal{D}_t;\boldsymbol{\psi}_t)}\Big[\mathrm{KL}\big[q(\boldsymbol{\phi}_t \mid \boldsymbol{\theta}_t^{z_t},\mathcal{D}_t;\boldsymbol{\lambda}_t)||p(\boldsymbol{\phi}_t|\boldsymbol{\theta}_t^{z_t})\big]\Big]$$

$$- \mathbb{E}_{q(z_t;\boldsymbol{\eta}_t)}\Big[\mathrm{KL}\big[q(\boldsymbol{\theta}_t^{z_t} \mid \mathcal{D}_t;\boldsymbol{\psi}_t)||p(\boldsymbol{\theta}_t^{z_t} \mid \mathcal{D}_{1:t-1})\big]\Big]$$

$$- \mathbb{E}_{q(\boldsymbol{\beta};\boldsymbol{\gamma})}\Big[\mathrm{KL}\big[q(z_t;\boldsymbol{\eta}_t)||p(z_t|\boldsymbol{\beta})\big]\Big] - \mathrm{KL}\big[q(\boldsymbol{\beta};\boldsymbol{\gamma})||p(\boldsymbol{\beta})\big]$$

$$\triangleq \mathcal{L}_{\mathrm{VC-BML}}(\boldsymbol{\lambda}_t, \boldsymbol{\psi}_t, \boldsymbol{\eta}_t, \boldsymbol{\gamma}). \tag{24}$$

The second inequality is obtained by applying Jensen's inequality to derive the lower bound of the conditional likelihood: $\log p(\mathcal{D}_t \mid z_t, \mathcal{D}_{1:t-1})$. The ELBO of our likelihood function is represented as $\mathcal{L}_{\mathrm{VC-BML}}(\boldsymbol{\lambda}_t, \boldsymbol{\psi}_t, \boldsymbol{\eta}_t, \boldsymbol{\gamma})$, and the variational distributions are optimized by maximizing this ELBO.

**Variational distribution of $\boldsymbol{\beta}$.** To obtain the optimal variational distribution of $\boldsymbol{\beta}_k$, for $=1, \ldots, K-1$, we only consider the terms related to $\boldsymbol{\beta}_k$ in the ELBO, which are given by:

$$\mathcal{F}\big(q(\boldsymbol{\beta}_k;\boldsymbol{\gamma}_k)\big) = \mathbb{E}_{q(z_t;\boldsymbol{\eta}_t)}\Big[\mathbb{E}_{q(\boldsymbol{\beta};\boldsymbol{\gamma})}\big[\log p(\boldsymbol{\beta}_k) + \log p(\boldsymbol{z}_t|\boldsymbol{\beta})\big]\Big] - \mathbb{E}_{q(\boldsymbol{\beta}_k;\boldsymbol{\gamma}_k)}\big[\log q(\boldsymbol{\beta}_k;\boldsymbol{\gamma}_k)\big]. \tag{25}$$

By taking the functional derivative of above equation with respective to $q(\boldsymbol{\beta}_k;\boldsymbol{\gamma}_k)$ and equate to zero:

$$\frac{\partial}{\partial q(\boldsymbol{\beta}_k;\boldsymbol{\gamma}_k)}\mathcal{F}\big(q(\boldsymbol{\beta}_k;\boldsymbol{\gamma}_k)\big)$$

$$= \int q(z_t;\boldsymbol{\eta}_t)q(\boldsymbol{\beta}_{\backslash k};\boldsymbol{\gamma}_{\backslash k})\Big[\frac{\partial}{\partial q(\boldsymbol{\beta}_k;\boldsymbol{\gamma}_k)}\int q(\boldsymbol{\beta}_k;\boldsymbol{\gamma}_k)\big[\log p(\boldsymbol{\beta}_k) + \log p(\boldsymbol{z}_t|\boldsymbol{\beta}) - \log q(\boldsymbol{\beta}_k;\boldsymbol{\gamma}_k)\big]\mathrm{d}\boldsymbol{\beta}_k\Big]\mathrm{d}\boldsymbol{\beta}_{\backslash k}\mathrm{d}z_t$$

$$= \int q(z_t;\boldsymbol{\eta}_t)q(\boldsymbol{\beta}_{\backslash k};\boldsymbol{\gamma}_{\backslash k})\Big[\log p(\boldsymbol{\beta}_k) + \log p(\boldsymbol{z}_t|\boldsymbol{\beta}) - \log q(\boldsymbol{\beta}_k;\boldsymbol{\gamma}_k) - 1\Big]\mathrm{d}\boldsymbol{\beta}_{\backslash k}\mathrm{d}z_t,$$

where $q(\boldsymbol{\beta}_{\backslash k};\boldsymbol{\gamma}_{\backslash k})$ is the variational distribution of $\boldsymbol{\beta}$ without $\boldsymbol{\beta}_k$, we can obtain the variational distribution of $\boldsymbol{\beta}_k$ as:

$$\log q(\boldsymbol{\beta}_k;\boldsymbol{\gamma}_k) \propto \mathbb{E}_{q(z_t;\boldsymbol{\eta}_t)}\Big[\mathbb{E}_{q(\boldsymbol{\beta}_{\backslash k};\boldsymbol{\gamma}_{\backslash k})}\big[\log p(\boldsymbol{\beta}_k) + \log p(\boldsymbol{z}_t|\boldsymbol{\beta})\big]\Big]$$

$$= \log p(\boldsymbol{\beta}_k) + \mathbb{E}_{q(z_t;\boldsymbol{\eta}_t)}\Big[\mathbb{E}_{q(\boldsymbol{\beta}_{\backslash k};\boldsymbol{\gamma}_{\backslash k})}\Big[\sum_{k=1}^{K-1}\Big(\boldsymbol{z}_{t,k}\log\boldsymbol{\beta}_k + \sum_{l=1}^{k-1}\boldsymbol{z}_{t,k}\log(1-\beta_l)\Big)$$

$$+ \sum_{l=1}^{K-1}z_{t,K}\log(1-\beta_l)\Big]\Big]$$

$$\propto \log p(\boldsymbol{\beta}_k) + \mathbb{E}_{q(z_t;\boldsymbol{\eta}_t)}\Big[\boldsymbol{z}_{t,k}\log\boldsymbol{\beta}_k + \sum_{r=k+1}^{K-1}\boldsymbol{z}_{t,r}\log(1-\boldsymbol{\beta}_k) + z_{t,K}\log(1-\boldsymbol{\beta}_k)\Big]$$

$$= \log p(\boldsymbol{\beta}_k) + \boldsymbol{\eta}_{t,k}\log\boldsymbol{\beta}_k + \sum_{r=k+1}^{K}\boldsymbol{\eta}_{t,r}\log(1-\boldsymbol{\beta}_k). \tag{26}$$

Since $p(\boldsymbol{\beta}_k) \sim \text{Beta}(\cdot; 1, \alpha)$ is a Beta distribution, we have:

$$\log p(\boldsymbol{\beta}_k) \propto (1 - 1) \log \boldsymbol{\beta}_k + (\alpha - 1) \log(1 - \boldsymbol{\beta}_k). \tag{27}$$

Consequently, the optimal variational distribution of $\boldsymbol{\beta}_k$ can be represented as:

$$\log q(\boldsymbol{\beta}_k; \boldsymbol{\gamma}_k) \propto \left(0 + \boldsymbol{\eta}_{t,k}\right) \log \boldsymbol{\beta}_k + \left(\alpha - 1 + \sum_{r=k+1}^{K} \boldsymbol{\eta}_{t,r}\right) \log(1 - \boldsymbol{\beta}_k), \tag{28}$$

which is also a beta distribution with parameters:

$$\boldsymbol{\gamma}_{k,1} = 1 + \boldsymbol{\eta}_{t,k}, \tag{29}$$

$$\boldsymbol{\gamma}_{k,2} = \alpha + \sum_{r=k+1}^{K} \boldsymbol{\eta}_{t,r}. \tag{30}$$

As a result, the optimal variatioal distribution of $\boldsymbol{\beta}_k$ is a beta distribution, which is represented as $q(\boldsymbol{\beta}_k) \sim \text{Beta}(\boldsymbol{\beta}_k; \gamma_{k,1}, \gamma_{k,2})$.

**Variational distribution of** $z_t$**.** Since $z_t$ is a category variable, we assume the variational distribution of $z_t$ to be a categorical distribution parameterized with $\boldsymbol{\eta}_t$:

$$q(z_t; \boldsymbol{\eta}_t) = \text{Categorical}_K(z_t; \boldsymbol{\eta}_t). \tag{31}$$

Similarly, to obtain the optimal variational distribution of $z_t$, we only consider the terms related to $z_t$ in the ELBO, which are given by:

$$\mathcal{F}\big(q(z_t; \boldsymbol{\eta}_t)\big) = \sum_{k=1}^{K} \boldsymbol{\eta}_{t,k} \cdot \mathbb{E}_{q(\boldsymbol{\beta}; \boldsymbol{\gamma})}\big[\log p(z_t = k \mid \boldsymbol{\beta})\big] - \boldsymbol{\eta}_{t,k} \log \boldsymbol{\eta}_{t,k}$$
$$- \boldsymbol{\eta}_{t,k} \cdot \text{KL}\big[q(\boldsymbol{\theta}_t^{z_t=k} \mid \mathcal{D}_t; \boldsymbol{\psi}_t) || p(\boldsymbol{\theta}_t^{z_t=k} \mid \mathcal{D}_{1:t-1})\big]$$
$$- \boldsymbol{\eta}_{t,k} \cdot \mathbb{E}_{q(\boldsymbol{\theta}_t^{z_t=k} \mid \mathcal{D}_t; \boldsymbol{\psi}_t)}\left[\text{KL}\big[q(\boldsymbol{\phi}_t \mid \boldsymbol{\theta}_t^{z_t=k}, \mathcal{D}_t; \boldsymbol{\lambda}_t) || p(\boldsymbol{\phi}_t | \boldsymbol{\theta}_t^{z_t=k})\big]\right]. \tag{32}$$

By taking derivative of $\mathcal{F}\big(q(z_t; \boldsymbol{\eta}_t)\big)$ with respective to $\boldsymbol{\eta}_{t,k}$ and equate to zero:

$$\frac{\partial}{\partial \boldsymbol{\eta}_{t,k}} \mathcal{F}\big(q(z_t; \boldsymbol{\eta}_t)\big) = \mathbb{E}_{q(\boldsymbol{\beta}; \boldsymbol{\gamma})}\big[\log p(z_t = k \mid \boldsymbol{\beta})\big] - \log \boldsymbol{\eta}_{t,k} + 1$$
$$- \text{KL}\big[q(\boldsymbol{\theta}_t^{z_t=k} \mid \mathcal{D}_t; \boldsymbol{\psi}_t) || p(\boldsymbol{\theta}_t^{z_t=k} \mid \mathcal{D}_{1:t-1})\big]$$
$$- \mathbb{E}_{q(\boldsymbol{\theta}_t^{z_t=k} \mid \mathcal{D}_t; \boldsymbol{\psi}_t)}\left[\text{KL}\big[q(\boldsymbol{\phi}_t \mid \boldsymbol{\theta}_t^{z_t=k}, \mathcal{D}_t; \boldsymbol{\lambda}_t) || p(\boldsymbol{\phi}_t | \boldsymbol{\theta}_t^{z_t=k})\big]\right], \tag{33}$$

we can obtain the optimal value for $\boldsymbol{\eta}_{t,k}$ as:

$$\log \boldsymbol{\eta}_{t,k} \propto \mathbb{E}_{q(\boldsymbol{\beta}; \boldsymbol{\gamma})}\big[\boldsymbol{\pi}_k\big] - \text{KL}\big[q(\boldsymbol{\theta}_t^{z_t=k} \mid \mathcal{D}_t; \boldsymbol{\psi}_t) || p(\boldsymbol{\theta}_t^{z_t=k} \mid \mathcal{D}_{1:t-1})\big]$$
$$- \mathbb{E}_{q(\boldsymbol{\theta}_t^{z_t=k} \mid \mathcal{D}_t; \boldsymbol{\psi}_t)}\left[\text{KL}\big[q(\boldsymbol{\phi}_t \mid \boldsymbol{\theta}_t^{z_t=k}, \mathcal{D}_t; \boldsymbol{\lambda}_t) || p(\boldsymbol{\phi}_t | \boldsymbol{\theta}_t^{z_t=k})\big]\right], \tag{34}$$

under the constraint that $\sum_{k=1}^{K} \boldsymbol{\eta}_{t,k} = 1$.

**Variational distribution of** $\boldsymbol{\theta}_t$ The meta-parameter distribution is assumed to contain a mixture of component distributions. Deriving the exact posterior distribution of meta-parameter is intractable due to the nonlinearities of the model. We follow the principle of variational inference and assume the variational distribution of meta-parameter for the $k$-th component to be:

$$q\left(\boldsymbol{\theta}_t^{z_t=k} \mid \mathcal{D}_t; \boldsymbol{\psi}_t\right) = \mathcal{N}\left(\boldsymbol{m}_t^k, \Lambda_t^k\right), \tag{35}$$

which is a Gaussian distribution with parameters $\boldsymbol{\psi}_t^{z_t=k} = \{\boldsymbol{m}_t^k, \Lambda_t^k\}$. $\boldsymbol{m}_t^k$ is the mean vector and $\Lambda_t^k$ is the semi-positive covariance matrix of the $k$-th component. Following the principle of variational continual learning [1], we assume that the prior of $\boldsymbol{\theta}_t^k$ is given by the variational posterior of $\boldsymbol{\theta}_{t-1}^k$, i.e., $\boldsymbol{\mu}_t^k = \boldsymbol{m}_{t-1}^k, \boldsymbol{\Sigma}_t^k = \boldsymbol{\Lambda}_{t-1}^k$. The variational distribution of $\boldsymbol{\theta}_t^k$ can be recursively updated by maximizing EBLO with stochastic back propagation and the reparameterization trick [2].

**Variational distribution of $\phi_t$**    For $q(\phi_t \mid \boldsymbol{\theta}_t^{z_t}, \mathcal{D}_t; \boldsymbol{\lambda}_t)$, $\phi_t$ are the weights of a deep neural network and $\boldsymbol{\lambda}_t$ are variational parameters of the weight distribution (i.e., means and standard deviations). Learning local variational parameters $\boldsymbol{\lambda}_t$ for high-dimension $\phi_t$ becomes difficult due to the costs of storing and computing $\boldsymbol{\lambda}_t$. Therefore, we compute $\boldsymbol{\lambda}_t$ with amortized variational inference (AVI) [3], where a global learned model is used to predict $\boldsymbol{\lambda}_t$ from the support set $\mathcal{D}_t^S$. It is shown that inference can be amortized by finding a good initialization, a la MAML [4]. Hence, from a global initialization $\boldsymbol{\theta}_t^{z_t=k}$, we produce the variational parameter $\boldsymbol{\lambda}_t$ by conducting several steps of gradient descents.

Let $\mathcal{L}_{\mathcal{D}_t^S}(\boldsymbol{\psi}_t, \boldsymbol{\lambda}_t) = -\mathbb{E}_{q(\phi_t \mid \boldsymbol{\theta}_t^{z_t}, \mathcal{D}_t; \boldsymbol{\lambda}_t)} \left[ \log p\left(\mathcal{D}_t^S \mid \phi_t\right) \right] + \mathrm{KL}\left[ q(\phi_t \mid \boldsymbol{\theta}_t^{z_t}, \mathcal{D}_t; \boldsymbol{\lambda}_t) \| p\left(\phi_t \mid \boldsymbol{\theta}_t^{z_t}\right) \right]$ be the loss on the dataset $\mathcal{D}_t^S$. We define the procedure of stochastic gradient descent, $SGD_J(\boldsymbol{\theta}_t^{z_t}, \mathcal{D}_t^S, \epsilon)$, to produce $\boldsymbol{\lambda}_t$ from the the global initialization $\boldsymbol{\theta}_t^{z_t=k}$:

$$
\begin{aligned}
&1.\ \boldsymbol{\lambda}_t^{(0)} = \boldsymbol{\theta}_t^{z_t}. \\
&2.\ \text{for } j = 0, \ldots, J-1, \text{ set} \\
&\quad \boldsymbol{\lambda}_t^{(j+1)} = \boldsymbol{\lambda}_t^{(j)} - \epsilon \nabla_{\boldsymbol{\lambda}_t^{(j)}} \mathcal{L}_{\mathcal{D}_t^S}\left(\boldsymbol{\psi_t}, \boldsymbol{\lambda}_t^{(j)}\right). \\
&3.\ \boldsymbol{\lambda}_t = \boldsymbol{\lambda}_t^{(J)}
\end{aligned}
\tag{36}
$$

$J$ is the number of steps of gradient descents and $\epsilon$ is the learning rate. Hereby $q(\phi_t \mid \boldsymbol{\theta}_t^{z_t}, \mathcal{D}_t; \boldsymbol{\lambda}_t) = q(\phi_t; SGD_J(\boldsymbol{\theta}_t^{z_t}, \mathcal{D}_t^S, \epsilon))$.

## D    Pseudo codes

We summarize the meta-training process of our VC-BML in algorithm 1. For simplicity, in the algorithm, we only use a single batch of data to update the parameters. It can be easily extended to mini-batch training by sampling a batch of data and aggregating the gradients as in [4].

---

**Algorithm 1:** Variational Continual Bayesian Meta-Learning.

**Input** : Task distribution $p(\tau)$ and data distribution $p(\mathcal{D} \mid \tau)$,
  maximum number of mixture components $K$,
  concentration parameter $\alpha$,
  number of inner update steps $J$,
  inner-update learning rate $\epsilon$,
  outer-update learning rate $\zeta$.

1 Initialize parameters: $\boldsymbol{\mu}_1^k, \boldsymbol{\Sigma}_1^k, \quad \forall k = 1, \cdots, K$;
2 **for** $t = 1, \ldots$ **do**
3   Sample a task $\tau_t \sim p(\tau)$;
4   Sample task dataset $\mathcal{D}_t = \{\mathcal{D}_t^S, \mathcal{D}_t^Q\} \sim p(\mathcal{D}_t \mid \tau_t)$;
5   Initialize $\boldsymbol{\eta}_{t,k}, \forall k = 1, \ldots, K$;
6   Update $\boldsymbol{\gamma}_k$ according to Equation 29 and Equation 30, $\forall k = 1, \ldots, K-1$ ;
7   // Initialize variational parameters $\boldsymbol{\psi}_t^k = \{\boldsymbol{m}_t^k, \boldsymbol{\Lambda}_t^k\}$
8   $\boldsymbol{m}_t^k \leftarrow \boldsymbol{\mu}_t^k, \forall k = 1, \ldots, K$;
9   $\boldsymbol{\Lambda}_t^k \leftarrow \boldsymbol{\Sigma}_t^k, \forall k = 1, \ldots, K$;
10   **while** *not converge* **do**
11    Update $\boldsymbol{\eta}_{t,k}$ according to Equation 34, $\forall k = 1, \ldots, K$;
12    **for** $k = 1, \ldots, K$ **do**
13     Sample $\boldsymbol{\theta}_t^{z_t=k}$ from $q(\boldsymbol{\theta}_t^{z_t=k} \mid \mathcal{D}_t; \boldsymbol{\psi}_t)$;
14     Update $\boldsymbol{\lambda}_t \leftarrow SGD_J(\boldsymbol{\theta}_t^{z_t=k}, \mathcal{D}_t^S, \epsilon)$;
15     Update $\boldsymbol{\psi}_t^k \leftarrow \boldsymbol{\psi}_t^k + \zeta \nabla_{\boldsymbol{\psi}_t^k} \mathcal{L}(\boldsymbol{\lambda}_t, \boldsymbol{\psi}_t, \boldsymbol{\eta}_t, \boldsymbol{\gamma})$;
16   // Update priors of meta-parameter distributions
17   $\boldsymbol{\mu}_{t+1}^k \leftarrow \boldsymbol{m}_t^k, \forall k = 1, \ldots, K$;
18   $\boldsymbol{\Sigma}_{t+1}^k \leftarrow \boldsymbol{\Lambda}_t^k, \forall k = 1, \ldots, K$ .

---

# E  Detailed Experimental Settings

## E.1  Experimental details about baselines

For fair comparison, we adopt the widely-used neural network architecture proposed by [5] in our VC-BML and all the baselines. We additionally describe the experimental settings of baselines.

**TOE:** We use the same Bayesian meta-learning model in our VC-BML as the base learner of TOE. The differences between TOE and VC-BML are that TOE only maintains a single meta-parameter distribution and is meta-trained on all the available data so far (i.e., $\mathcal{D}_{1:t}$) at each timestamp $t$. We train TOE with 3 batch gradient descent steps with a learning rate of $0.1$ for inner update, and use Adam optimizer with a learning rate of $0.001$ for outer update.

**TFS:** The experimental settings of TFS is the same as TOE, and the difference is that TFS is only meta-trained on the current data (i.e., $\mathcal{D}_t$) at each timestamp $t$.

**FTML** [6]: We apply a slight modification to the original FTML algorithm. FTML utilizes all the available data so far (i.e., $\mathcal{D}_{1:t}$) for meta-training, which is memory consuming and contradicting to our continual learning setting of streaming data and tasks. For fair comparison, we only meta-train FTML on the currently available data (i.e., $\mathcal{D}_t$), which is the same as our VC-BML. Moreover, we evaluate FTML on the unseen tasks (i.e., tasks sampled from meta-test set) instead of the training tasks that the original FTML used. The rest of experimental settings remain unchanged compared to the original FTML paper. Throughout the experiments, we train FTML with 3 inner gradient descent steps and a inner learning rate of $0.1$, and train the convolution network using Adam optimizer with a learning rate of $0.001$. In the setting of non-stationary task distribution, we meta-train FTML for $5000$ steps on each dataset. Classification accuracies are calculated from the meta-test sets of each dataset. Specifically, to evaluate FTML, we randomly sample 100 tasks from the meta-test sets and calculate the average classification accuracies.

**OSML** [7]: In the original paper, the authors use a well pre-trained convolution network to initialize their OSML model. However, both our VC-BML and other baselines are randomly initialized and trained from scratch. It would be unfair to adopt the original initialization procedure in OSML. Consequently, in our experiments, we randomly initialize the parameters of OSML and train OSML from scratch. The experimental setting of OSML is similar to FTML. Throughout the experiments, we use 3 knowledge blocks in each layer of the convolution network. In the settings of non-stationary task distribution, we use 5 inner gradient descent steps with an inner learning rate of $0.01$, and use an outer learning rate of $0.001$ on Omniglot, $0.0001$ on CIFAR-FS, *mini*Imagenet and VGG-Flowers.

**DPMM** [8]: The experimental settings of DPMM is also similar to FTML. In the settings of non-stationary task distribution, we use an inner learning rate of $0.1$, an outer learning rate of $0.001$ on Omniglot and CIFAR-FS, and use an inner learning rate of $0.01$, an outer learning rate of $0.0001$ on *mini*Imagenet and VGG-Flowers.

**OSAKA** [9]: In the original setting of OSAKA, it involves two learning stages: pre-training stage for parameter initialization and continual learning stage for online adaption. Similar to OSML, for fair comparison, we omit the pre-training stage and only deploy the continual learning stage in our experiments. To learn OSAKA, we use 3 inner gradient descent steps with an inner learning rate of $0.1$, and use an outer learning rate of $0.001$ on Omniglot, CIFAR-FS, $0.0001$ on *mini*Imagenet, VGG-Flowers. We train OSAKA for $5000$ steps with Adam optimizer on each dataset.

**BOMVI** [10]: In our experiments, we use variational inference to approximate the posterior of meta-parameters. We train BOMVI with 5 batch gradient descent steps with a learning rate of $0.1$ for inner update, and use Adam optimizer with a learning rate of $0.001$ for outer update. The other settings are also similar to our VC-BML.

The hyperparameters of these baselines are chosen in a similar way as our VC-BML (see Section E.3.2 for details) and are tuned to be optimal based on their performance on the validation sets.

## E.2  Practical Implementation Details

When inferring posterior of $z$ (Equation 34), the likelihood of task dataset, i.e., $p(\mathcal{D}_t \mid \phi)$, is absent from the equation. We empirically found that it is beneficial to incorporate the likelihood function into the equation. Practically, the likelihood of task dataset is leveraged to calculate the posterior of $z$,

Table 1: The CNN architecture for VC-BML and baselines.

| Output size | Layers |
| --- | --- |
| $28 \times 28 \times 3$ | Input images |
| $14 \times 14 \times 64$ | *conv*2d($3 \times 3$, stride=1, padding=1), BatchNormalization, Relu, *maxPool*($2 \times 2$, stride=2) |
| $7 \times 7 \times 64$ | *conv*2d($3 \times 3$, stride=1, padding=1), BatchNormalization, Relu, *maxPool*($2 \times 2$, stride=2) |
| $3 \times 3 \times 64$ | *conv*2d($3 \times 3$, stride=1, padding=1), BatchNormalization, Relu, *maxPool*($2 \times 2$, stride=2) |
| $1 \times 1 \times 64$ | *conv*2d($3 \times 3$, stride=1, padding=1), BatchNormalization, Relu, *maxPool*($2 \times 2$, stride=2) |
| 64 | flatten |

which is given by:

$$
\begin{aligned}
\log \boldsymbol{\eta}_{t,k} \propto & \mathbb{E}_{q(\boldsymbol{\beta};\boldsymbol{\gamma})}\big[\boldsymbol{\pi}_k\big] - \mathrm{KL}\big[q(\boldsymbol{\theta}_t^{z_t=k} \mid \mathcal{D}_t; \boldsymbol{\psi}_t)||p(\boldsymbol{\theta}_t^{z_t=k} \mid \mathcal{D}_{1:t-1})\big] \\
& - \mathbb{E}_{q(\boldsymbol{\theta}_t^{z_t=k}|\mathcal{D}_t;\boldsymbol{\psi}_t)}\bigg[\mathrm{KL}\big[q(\boldsymbol{\phi}_t \mid \boldsymbol{\theta}_t^{z_t=k}, \mathcal{D}_t; \boldsymbol{\lambda}_t)||p(\boldsymbol{\phi}_t|\boldsymbol{\theta}_t^{z_t=k})\big]\bigg] \\
& + \mathbb{E}_{q(z_t;\boldsymbol{\eta}_t)q(\boldsymbol{\theta}_t^{z_t}|\mathcal{D}_t;\boldsymbol{\psi}_t)q(\boldsymbol{\phi}_t|\boldsymbol{\theta}_t^{z_t},\mathcal{D}_t;\boldsymbol{\lambda}_t)}\bigg[\log p(\mathcal{D}_t \mid \boldsymbol{\phi}_t)\bigg].
\end{aligned}
\tag{37}
$$

### E.3 Non-stationary task distribution

#### E.3.1 Datasets

**Omniglot** [11]: Omniglot contains 1623 handwritten characters (classes), each with 20 examples. All characters are grouped in 50 alphabets. We follow previous work [5] and randomly split the dataset: 1100 characters for meta-training, 100 characters for validation the remaining 423 characters for meta-test. All images are resized to $28 \times 28$. Since we study the 5-way 5-shot classification problem, we form a sequence of tasks by sampling 5 classes from the meta-training set with replacement as a task. For each task, we randomly sample 5 examples for each class in the support set and 15 examples for each class in the query set. The same sampling strategy of support set and query set is also used in validation and testing.

**CIFAR-FS** [12]: CIFAR-FS is adapted from the CIFAR-100 dataset [13] for few-shot learning. It contains 100 classes of images, each with 600 instances. We use the same split as [12]: 64 classes for meta-training, 16 classes for validation and 20 classes for meta-test.

*mini***Imagenet**: The *mini*Imagenet is a challenging dataset constructed from ImageNet [14] , which comprises a total of 100 different classes, each with 600 instances. We use the same splits of [15], where there are 64 classes for meta-training, 16 classes for validation and 20 classes for meta-test.

**VGG-Flowers**: The VGG-Flowers [16] contains 102 different classes of flowers, with 8,189 instances in total. We randomly sample 66 classes for meta-training, 16 classes for validation and 20 classes for meta-test.

Note that on CIFAR-FS, *mini*Imagenet and VGG-Flowers datasets, we follow the same preprocessing steps as Omniglot to generate a sequence of tasks. Moreover, the datasets we are using are publicly available image data with their licenses, which have been frequently used in the research community, and they do not contain any personally identifiable information or offensive content.

#### E.3.2 Settings

As the latent variables in this paper are meta-parameters and task-specific parameters, the dimensionality of the latent space is actually determined by the number of parameters in the deep neural network. In particular, we define a CNN architecture and present its details in Table 1. Roughly, there are 112,000 parameters in the defined CNN. So the dimensionality of the latent space is about 112,000.

For each task, we randomly sample 5 and 15 shots per class to obtain $\mathcal{D}_t^S$ and $\mathcal{D}_t^Q$ respectively. We use the same convolution network architecture proposed by [5] as our base model, which is

Table 2: Hyperparameters for VC-BML on the settings of non-stationary task distribution.

| Hyperparameter | Omniglot | CIFAR-FS | *mini*Imagenet | VGG-Flowers |
|---|---|---|---|---|
| Number of inner SGD steps | 1 | 5 | 3 | 3 |
| Inner-update learning rate | 0.1 | 0.1 | 0.1 | 0.05 |
| Down-weighting parameters of KL-divergence $\nu$ | 0.05 | 0.005 | 1.0 | 1.0 |
| Number of outer SGD steps | 5000 | 5000 | 5000 | 5000 |
| Outer-update learning rate | 0.001 | 0.001 | 0.0001 | 0.0001 |

described in Table 1. After the convolution layers, we use a fully-connected layer to predict the class probabilities with softmax function. For fair comparison, the same neural network architecture is used in all the baselines. Moreover, since we aim to deal with streaming tasks, at each time step $t$, we meta-train VC-BML and baselines (except TOE) on the current task.

To train different datasets on the same network, we resize the images from these datasets to $28 \times 28 \times 3$. We set the maximum number of mixture components $K = 6$ and the concentration parameter $\alpha = 1$. Throughout the experiments, we use batch gradient descent to obtain the task-specific parameters and Adam optimizer to update the variational parameters with a batch size of 32. We use 5 samples for Monte Carlo estimation.

For other hyperparameters, we choose the inner update steps from $\{1, 3, 5\}$, inner update learning from $\{0.001, 0.01, 0.05, 0.1\}$, outer update steps from $\{2500, 5000\}$, outer update learning rate from $\{0.0001, 0.001, 0.01\}$ and $\nu$ from $\{0.005, 0.05, 0.01, 0.1, 1.0\}$. We choose hyperparameters based on the evaluation scores on the validation sets for both our VC-BML and baselines. For evaluation, we randomly sample 100 tasks from the meta-test set of each dataset, perform 5 inner gradient descent steps to obtain task-specific parameters, and calculate the mean classification accuracies. Additional hyperparameter settings can be found in Table 2. We ran experiments for 5 times on each setting with different random seed, and report their average accuracies and standard deviations.

We implement our model based on the code of BOMVI [10]. We appreciate the authors for making their code publicly available. We ran experiments on a single machine with 8 NVIDIA GeForce RTX 2080Ti with 11GB memory, 56 Intel Xeon CPUs (E5-2680 v4 @ 2.40GHz). It takes about 80 hours to train VC-BML with K=6 on each dataset.

## F    Additional Experimental results

### F.1    Meta-test Accuracies on each Dataset at Different Meta-training Stage

Due to the space limit, in the main text, we only report the meta-test accuracies on each dataset at the beginning and ending of meta-training stages. We additionally report the full results in this section, which is presented in Figure 3. In the main text, we observe that parametric methods suffer from more severer knowledge forgetting problem than Bayesian methods on Omniglot dataset. From Figure 3, we can observe that similar results can also be found on CIFAR-FS and *mini*Imagenet datasets. For example, on CIFAR-FS, the accuracies of parametric methods (FTML, OSML, DPMM and OSAKA) decrease from 69.79%-71.41% to 52.29%-59.74% at the end of training, while the accuracies of Bayesian methods (BOMVI, VC-BML) barely change (from 69.12%-75.61% to 70.16%-75.20%). Similarly, on *mini*Imagenet, the accuracies of parametric methods decrease from 55.85%-59.17% to 41.43%-44.93% at the end of training, while the accuracies of Bayesian methods still barely change (from 58.38%-61.24% to 57.94%-61.43%). Moreover, in the main text, we observe that Bayesian methods have better generalization capability on unseen tasks than parametric methods when meta-trained on Omniglot dataset. Similar results can also be found on CIFAR-FS and *mini*Imagenet datasets. It can be observed from Figure 3 that Bayesian methods, i.e., VC-BML and BOMVI, always obtain the best and the second best performance on unseen tasks, i.e., *mini*Imagenet and VGG-Flowers at CIFAR-FS meta-training stage and VGG-Flowers at *mini*Imagenet meta-training stage. These results validate the analyses in the main text.

Table 3: Meta-test accuracies (%) on each dataset at different meta-training stage. "Average" represents the mean meta-test accuracies on four datasets. (The best performance per dataset per meta-training stage is marked with boldface and the second best is marked with underline.)

| Meta-training Stage | Algorithms | Omniglot | CIFAR-FS | *mini*Imagenet | VGG-Flowers | Average |
|---|---|---|---|---|---|---|
| **Omniglot** | TOE | 98.43 ± 0.48 | 42.83 ± 1.77 | 33.09 ± 1.44 | 64.19 ± 2.11 | 59.64 ± 1.45 |
| | TFS | 98.22 ± 0.50 | 43.36 ± 1.91 | 33.49 ± 1.38 | 63.18 ± 2.05 | 59.56 ± 1.46 |
| | FTML | 99.11 ± 0.16 | 42.48 ± 1.88 | 34.69 ± 1.22 | 57.32 ± 2.06 | 58.40 ± 1.33 |
| | OSML | 99.30 ± 0.29 | 41.20 ± 1.82 | 32.52 ±1.33 | 55.39 ± 2.19 | 57.10 ± 1.41 |
| | DPMM | **99.33 ± 0.28** | 42.31 ± 1.62 | 34.40 ± 1.61 | 57.58 ±2.16 | 58.45 ± 1.42 |
| | OSAKA | 99.01 ± 0.29 | 43.24 ± 1.79 | 35.44 ± 1.35 | 60.13 ± 2.08 | 59.46 ± 1.38 |
| | BOMVI | 96.25 ± 0.66 | 48.22 ±1.94 | 37.83 ±1.45 | 66.24 ±1.98 | 63.06 ± 1.51 |
| | VC-BML | 98.62 ± 0.38 | **50.86 ± 1.94** | **38.11 ± 1.51** | **68.63 ± 2.08** | **64.06 ± 1.48** |
| **CIFAR-FS** | TOE | 96.07 ± 0.96 | 65.03 ± 1.82 | 53.32 ± 1.61 | 73.33 ±1.64 | 71.94 ± 1.51 |
| | TFS | 96.27 ± 0.68 | 68.51 ± 1.91 | 54.77 ±1.70 | 75.78 ± 1.69 | 73.83 ± 1.50 |
| | FTML | 94.87 ±0.80 | 69.79 ± 1.82 | 54.81 ± 1.67 | 73.33 ± 1.81 | 73.20 ± 1.53 |
| | OSML | 97.71 ± 0.69 | 70.11 ± 1.69 | 54.91 ± 1.75 | 76.58 ± 1.99 | 74.83 ± 1.53 |
| | DPMM | 95.63 ± 0.58 | 71.41 ± 1.84 | 52.74 ± 1.70 | 71.08 ± 1.99 | 72.71 ± 1.52 |
| | OSAKA | 95.36 ± 0.72 | 70.38 ± 1.76 | 54.57 ± 1.62 | 72.16 ± 1.96 | 73.12 ± 1.52 |
| | BOMVI | 96.98 ± 0.63 | 69.12 ± 1.86 | 55.59 ± 1.61 | 77.15 ± 1.85 | 74.71 ± 1.49 |
| | VC-BML | **98.28 ± 0.48** | **75.61 ± 1.82** | **60.15 ± 1.68** | **80.63 ± 1.73** | **78.67 ± 1.43** |
| *mini*Imagenet | TOE | 93.24 ± 1.07 | 58.37 ± 1.88 | 51.04 ± 1.64 | 71.90 ± 1.93 | 68.64 ± 1.63 |
| | TFS | 95.21 ± 1.11 | 63.75 ± 2.02 | 55.92 ± 1.67 | 73.09 ± 1.82 | 71.99 ± 1.66 |
| | FTML | 93.00 ± 0.85 | 67.05 ± 1.77 | 55.85 ± 1.60 | 71.74 ± 1.96 | 71.91 ± 1.55 |
| | OSML | 96.36 ± 0.53 | 69.87 ± 1.95 | 59.17 ± 1.59 | 77.74 ± 1.64 | 75.79 ± 1.43 |
| | DPMM | 94.11 ± 0.72 | 69.85 ± 1.77 | 57.92 ± 1.61 | 72.16 ± 1.66 | 73.51 ± 1.44 |
| | OSAKA | 95.84 ± 0.69 | 70.34 ± 1.89 | 56.83 ± 1.86 | 75.44 ± 1.73 | 74.61 ± 1.54 |
| | BOMVI | 96.76 ± 0.74 | 69.81 ± 1.84 | 58.38 ± 1.65 | 78.23 ± 1.59 | 75.80 ± 1.46 |
| | VC-BML | **98.23 ± 0.39** | **75.40 ± 1.99** | **61.24 ± 1.64** | **82.13 ± 1.84** | **79.25 ± 1.46** |
| **VGG-Flowers** | TOE | 93.12 ± 1.18 | 50.07 ± 2.20 | 41.97 ± 1.74 | 73.13 ± 1.71 | 64.57 ± 1.71 |
| | TFS | 87.84 ± 1.89 | 51.99 ± 1.89 | 41.53 ± 1.74 | 80.62 ± 1.41 | 65.50 ± 1.73 |
| | FTML | 87.01 ± 1.35 | 52.29 ± 1.90 | 41.43 ± 1.67 | 81.10 ± 1.58 | 65.46 ± 1.63 |
| | OSML | 93.14 ± 0.90 | 59.74 ± 1.85 | 48.44 ± 1.60 | 82.25 ± 1.43 | 70.89 ± 1.44 |
| | DPMM | 92.71 ± 0.76 | 58.76 ±1.83 | 46.95 ± 1.54 | 81.92 ± 1.48 | 70.09 ± 1.40 |
| | OSAKA | 90.05 ± 1.13 | 53.37 ± 2.01 | 44.93 ± 1.68 | 80.37 ± 1.73 | 67.18 ± 1.64 |
| | BOMVI | 96.60 ± 0.55 | 70.16 ± 1.84 | 57.94 ± 1.65 | 81.54 ± 1.81 | 76.56 ± 1.46 |
| | VC-BML | **97.47 ± 0.73** | **75.20 ± 1.80** | **61.43 ± 1.49** | **83.39 ± 1.73** | **79.37 ±1.44** |

## F.2 Performance of VC-BML under Different Meta-training Order

In the setting of non-stationary task distribution, we sequentially meta-train VC-BML on four datasets: Omniglot, CIFAR-FS, *mini*Imagenet and VGG-Flowers. To show that the performance of VC-BML is irrelevant to the meta-training orders, we conduct further experiments by meta-training VC-BML and baselines under different orders of datasets. Specifically, we sequentially meta-train VC-BML and baselines on: VGG-Flowers, *mini*Imagenet, CIFAR-FS and Omniglot, and show the experimental results in Figure 1. It can be observed that the parametric baselines (FTML, DPMM, OSML and OSAKA) still suffer from the forgetting problem when the task distribution shifts, while our VC-BML is able to alleviate this forgetting problem and maintains the performance when meta-trained on different task distributions. These results are in line with the results in the main text, which validates that VC-BML is agnostic to the meta-training orders, and that VC-BML is effective to handle the potentially dissimilar tasks from a non-stationary distribution, regardless of the meta-training orders.

## F.3 Number of Mixture Components

In the main text, when studying the effects of the number of mixture components $K$, we report the mean meta-test accuracies on all the learned dataset. We additional present the evolution of meta-test accuracies for VC-BML with different $K$ when meta-trained on sequentially arriving datasets, which is shown in Figure 2. The result shows that, after training on the four datasets, VC-BML with $K = 1$ has the worst performance, indicating the inefficiency of maintaining a single set of meta-parameters when encountering non-stationary task distribution. Moreover, the performance of VC-BML improves as K gets larger, suggesting that it is beneficial to instantiate disparate meta-parameter distributions for different task clusters. The similar performance between VC-BML with K=6 and VC-BML with K=4 indicates that it is not helpful to instantiate more meta-parameter distributions. The results and analyses coincide with the results and analyses in the section 7.3 of main text.

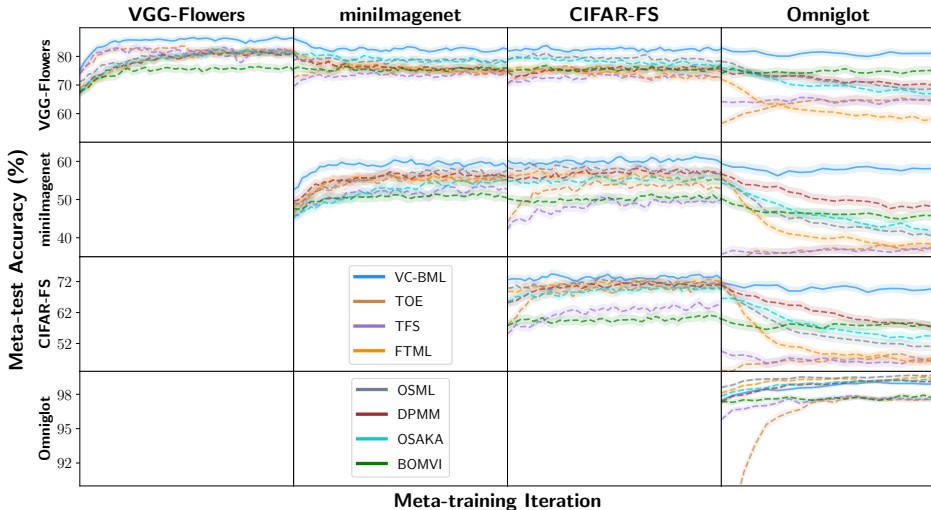

Figure 1: The evolution of the meta-test accuracies (%) for VC-BML and baselines when sequentially meta-trained on four datasets with different order. Each row represents the meta-test accuracies for a specific dataset over cumulative number of meta-training epochs.

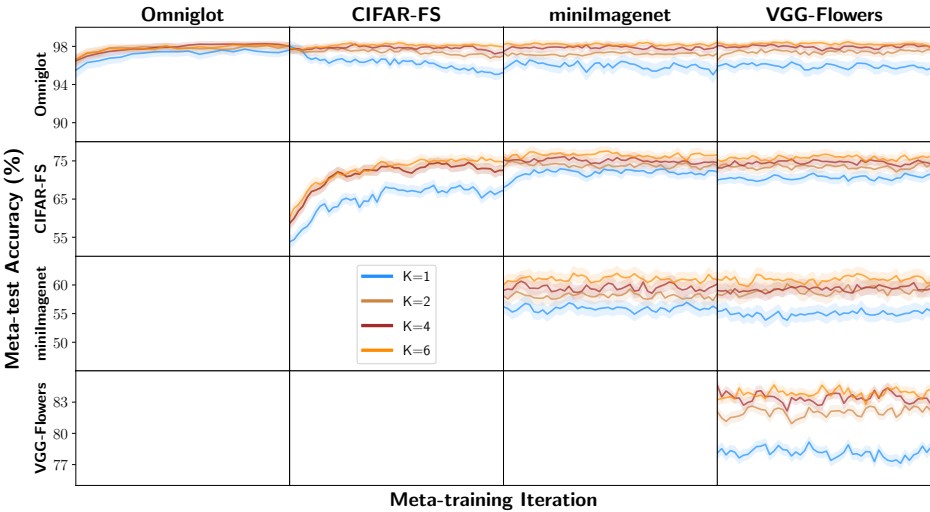

Figure 2: The evolution of the meta-test accuracies (%) for VC-BML with different $K$ when sequentially meta-trained on different datasets. Each row represents the meta-test accuracies for a specific dataset over cumulative number of meta-training epochs.

## F.4 Impact of KL-divergence

As mentioned in the main text, we suspect the reason why VC-BML does not obtain the best performance on Omniglot is the over regularization caused by the KL-divergence in the ELBO. To understand the impact of KL-divergence, we vary $\nu$, i.e., the down-wighting hyperparameter of KL-divergence, from $10^{-4} \sim 1.0$ on Omniglot dataset and the experimental results are shown in Figure 3. Note that we only compare against the results of FTML, OSML and DPMM because they have the best performance on Omniglot. The result shows that the performance of VC-BML is better than or comparative with baselines when $\nu$ is smaller than $0.01$, and then the performance decreases as $\nu$ gets larger. This confirms our hypothesis that the sub-optimal performance of VC-BML on Omniglot is caused by the over regularization of KL-divergence. It also motivates us to introduce $\nu$ to tune the KL-divergence to avoid over-fitting.

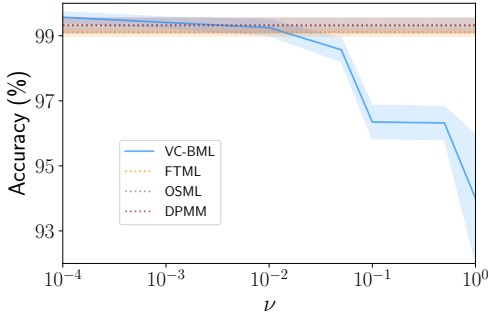

Figure 3: Classification accuracies of VC-BML and baselines under different $\nu$ on the Omniglot dataset.

Mode collapse: If the initial distribution is not adequately diverse, our model may not be able to capture all data modes. In order to reduce the strong loss caused by the data of collapsed modes, there are two methods. First, our model will adaptively add new components to the mixture distributions, and the new components will focus on collapsed modes to reduce relevant losses. Second, we can adjust the ELBO function to avoid the posterior collapsing to the prior. Specifically, we can use the hyperparameter to down-weight the KL-divergence term in ELBO. Smaller $\nu$ encourages our model to find the mode of the true posterior of the incoming data instead of collapsing to the prior.