# OpenReview forum: "Variational Continual Bayesian Meta-Learning"
_NeurIPS.cc/2021/Conference — NeurIPS 2021 Poster_

### Official Review · Reviewer_AgJH · 2021-07-12

**Rating:** 7
**Confidence:** 4

**Summary:**

This article presents a fully variational method to address the continual learning problem: training a neural network sequentially on different tasks, without forgetting preceding tasks.

The theoretical framework, namely variational inference with hierarchical random variables, is classical, which makes the general idea easy to understand.

The experiments are sound, concurrent methods have been replicated and tested (including naive baselines), and the main potential flaws or limitations have been discussed and addressed.

**Limitations And Societal Impact:**

As mentioned in the main review, potential limitations have been addressed.

**Main Review:**

## Detailed summary

The method is based on a hierarchical structure of random variables managing the final weights of the neural network to train. In order to deal with an unknown number of tasks, the weights $\phi$ of the NN depend on a categorical random variable $z$, whose range of values increases during training. In a nutshell, the range of $z$ is supposed to increase when the number and the variety of tasks increases.

The main theoretical difficulty is deriving the update rules for all variational parameters.

## Experiments

Setup: train sequentially the model on 4 given tasks (image classification), in a given order.

The authors compare their method to 7 other ones, including 2 naive baselines, which is sufficient to evaluate the method. They compare the results on each (seen) dataset during the sequential training on each task. The difficulty consists in keeping a good performance on a dataset when the model is being trained on another dataset. The proposed methods dominates all concurrent methods, except when learning the first task (first phase in Fig. 3).

Appendix F.2 is very important to check the fairness of the choice of the order of tasks in Section 5: the authors rerun the same experiments with the same tasks, *in inverse order*. The proposed method dominates all concurrent methods for each task, at any moment of the training. So we can reasonably be confident in the results presented in Fig. 3.

There is no comparison of training time between methods.

Section 5.2 confirms the intuition behind the range of the variable $z$: $z$ takes as many values as the number of tasks to perform. However, to be rigorous, Figure 4 should be replicated with 3 and 2 tasks, in order to check that the number of values $z$ can take are respectively 3 and 2.

## Questions

Why are results with OSAKA on Omniglot better in this paper than in the original one?

Is data augmentation used? It may be the cause of the impact of the KL mentioned in Section 5.3.

What about RNNs? For instance, text prediction (which is a simple task).

## Clarity

There is no issue of clarity if the reader is already familiar with hierarchical variational inference. Otherwise, the article may be hard to understand. For instance, the name of Section 4.2 could be changed to mention the update of variational parameters, which is the object of the section.

## Significance

The proposed method is an expected improvement in a field of medium importance (continual learning).

## Rebuttal

I am satisfied with the authors' answers. I maintain my rating, with a higher confidence.

**Time Spent Reviewing:**

6

---

> ### Author Response · Authors · 2021-08-09
> **Clarify questions about experiments**
>
> We thank the reviewer for the detailed comments. The following is our responses to the questions mentioned in the comments.
>
> 1. We provide the computational time of our model at lines 174-176 in Appendix, and will add this information for all baselines in the final version.
>
>      We agree that further experiments should be conducted to verify the effect of variable $z$. We will replicate our experiments on 2 and 3 tasks to validate that our method can find the actual number of task clusters in the final version.
>
>
> 2. The results of OSAKA on Omniglot is better than the original one because the experimental set-ups of this paper differ from the original OSAKA paper in the following two aspects: (1) The original OSAKA paper studies 10-way 1-shot classification tasks on Omniglot while this paper studies 5-way 5-shot classification tasks on Omniglot, which is a common task for evaluating the performance of meta-learning models. Since the tasks in our paper are easier than those in the original OSAKA paper, the performance of OSAKA in our paper is much higher than the original paper. (2) The OSAKA paper requires pre-training the model from some context. For fair comparison, in our paper we train all models, including OSAKA, from scratch. This might also lead to the performance differences. For more details about the experimental settings of OSAKA in our paper, please see Section E.1 of Appendix.
>
> 3. We didn’t use any data augmentation techniques.
>
> 4. The authors are not completely sure of the question. We guess the reviewer suggests an experiment of using RNN as a basic neural network for streaming text prediction tasks. We basically test the proposed model against baselines in image classification tasks, which is commonly adopted by the continual learning and meta-learning community. An additional experiment on RNN will be a good supplement.

---

### Official Review · Reviewer_h2Cd · 2021-07-15

**Rating:** 6
**Confidence:** 3

**Summary:**

**Update after author response:** I have raised my score from 4 to 6, even leaning towards 7 (but I would need to see the revised manuscript to confidently give a 7, which is unfortunately not possible, because a fair amount of my criticism is aimed at various passages throughout the paper and the overall claims and style of presenting them).

The paper addresses task agnostic continual (meta-) learning from a Bayesian perspective. The setting in continual learning is to learn a sequence of (potentially quite different) tasks, image classification tasks in the paper - the main difficulty is to learn novel tasks while avoiding catastrophic forgetting of previously seen tasks and negative interference (i.e. decreased learning or suboptimal solutions) from previously learned solutions. As discussed in the existing literature, a principled approach to do this is to perform Bayesian inference over the parameter-posterior given the current experience. This posterior can act as a prior for new experience - the standard sequential Bayesian posterior update. In practice, this update is intractable and the paper resorts to approximate inference: in essence a pior distribution is learned per cluster of related tasks. These task-group priors allow to rapidly adapt to each task within the cluster (which is performed in the paper in a MAML like fashion, i.e. via gradient update steps starting from an initial parameter-draw from the task-group pior). The main idea is thus to use a mixture distribution (mixture over Gaussians in the paper) to represent a multi-modal approximate posterior over task-parameters, which acts as a “good” prior for new observations, and thus allows for knowledge transfer between tasks of the same cluster. Since the number of clusters is not known in advance, and, more importantly, needs to grow with an increasing number of tasks observed the model employs a non-parametric posterior approximation (a Dirichlet process that models the number of Gaussian mixture components). The method is compared, and performs well, against a number of previously proposed methods.

**Contributions:**

1) Extension of the model used by a previous method [4] by using a mixture of Gaussians for the approximate posterior - each mixture component is supposed to model one cluster of tasks (in the previous paper each mixture component is a delta-distribution, leading to a Dirichlet Process Mixture Model DPMM). The number of mixture components is inferred from the data via a Dirichlet process (leading to a Dirichlet Process Gaussian Mixture Model DPGMM) - in contrast to the paper’s claims this has been done in precisely the same fashion in [4] (see Section 5 in that paper, which also uses the CRP construction). Significance: the improvement in the model is sensible but fairly incremental - it would have been interesting to allow for multi-modality on this level as well. Results are in favor of the improvement, but do not convincingly demonstrate that the observed improvements are solely caused by this modification (see comments below).

2) In contrast to [4] (which derives a MAP inference scheme), a variational inference scheme is derived. The innermost step of that scheme (MAML-type optimization using a few gradient-descent steps) relies on meta-learned amortized variational inference, variational inference of \theta relies on ELBO-type optimization via SGD and the reparametrization trick. Significance: the VI scheme is interesting and shown to scale to neural networks. Given the model complexity this is not trivial - however this complexity introduces many hyper-parameters and application of the model to new data might not be straightforward (the VI can easily break in subtle ways that are hard to diagnose and debug). Since the paper does not report in detail how most hyperparameters were found, and how sensitive the results are w.r.t. their choice becomes even more difficult.

3) Experiments on a sequence of four image classification tasks with comparison against two baselines and several related models from the literature. Results suggest that the model outperforms all other models in most cases. Significance: results look indeed promising, but at the current stage it is hard to attribute the observed gains in performance to the various improvements made. Important control experiments and ablations are missing to make sure the observed gains are actually due to the hypothetical claims that went into the model and inference improvements (more on this, see Improvements below).

**Ethical Concerns:**

No ethical concerns.

**Limitations And Societal Impact:**

Some limitations of the method are discussed. What is missing is how to set the hyperparameters (annealing schedule, prior-parameters, etc.) in practice and how brittle the VI scheme is. Additionally current experiments show results for a small number of tasks, I’d expect difficulties with inference for larger numbers of tasks, but it’s unclear whether that problem becomes apparent for slightly larger numbers of tasks or whether the method is reliable for dozens of tasks.

**Main Review:**

**Originality, Quality, Clarity, Correctness:**

While the use of a Dirichlet-process-Gaussian-mixture-model is certainly not novel, and fairly straightforward from a Bayesian viewpoint on continual learning, I have not seen this particular instantiation of the model in the continual learning literature before (though I might have some gaps, hence my lower confidence score). Getting VI to work reliably for such a model (with high-dimensional neural network weight-matrices as the parameters at the bottom of the hierarchical model) is no easy feat, and results in the paper seem to suggest that this works successfully. Unfortunately some of the claims in the paper are overstated and/or not sufficiently backed up by experimental evaluation - perhaps most importantly the differences to the model in [4] are more incremental than the paper claims (see criticism below for details). The clarity of the writing could be improved (see comments), and while there is no part that is clearly wrong in the paper the correctness of the interpretation of the results is unclear since important ablations and control experiments are missing (to show that the observed performance gains are due to the reasons claimed).

**Verdict:**

I think the main ingredients for a good paper are there - the central idea is a small, but sensible improvement in the model and the VI scheme looks interesting. Results shown in the paper are promising, but now extra work needs to be put in to make sure that the observed results come from the main innovation and to disentangle which innovation contributes to the performance gains to which degree. If done thoroughly, I think the paper has the potential for a fairly significant impact in the continual learning community. At its current stage I personally think that the paper does not have sufficient rigor and experimental evidence to back up all the claims made. I therefore suggest a major revision, and want to strongly encourage the authors to “go the extra mile” and turn this into the strongest paper possible, rather than rushing with publishing preliminary findings. I am happy to reconsider my verdict of course in light of the other reviews and the authors’ response.

After re-reading my review I want to point out that I’m not an author of [4] (or collaborator of the authors) - my aim is not to “defend” [4], but the differences need to made clear (both formally and experimentally).

**Pros:**

 * Clearly articulated, focussed problem setting. Main model improvement is sensible.
 * Derivation of a fairly involved VI scheme.
 * Promising results

**Cons:**

 * Overstated (or wrong) claims w.r.t. previous work - most importantly [4] in the paper uses a Dirichlet process to automatically infer (and increase) the number of mixture components.
 * Unclear whether the improved results (w.r.t. [4]) come from:
   * the Gaussian mixture model
   * the VI scheme as opposed to MAP inference in 4
   * the annealing of the regularizer
   * potentially more/better hyper-parameter tuning for the main method but not for the competitors
 * Some parts of the writing could be more rigorous (being precise about what exactly is more or less ‘Bayesian’, clearly describing what meta-parameters refers to, the params of the whole DPGMM?, the params of the Gaussian mixture?, the distr. over params induced by a single mixture component?, …)

**Criticism w.r.t. specific claims**

*Claimed advantages of the method*

 * L 221:  “the meta-parameters in VC-BML are represented by a dynamically updated mixture model compared to a single static distribution in [4].” - this claim is wrong, Section 5 in [4] uses the same CRP construction to dynamically adjust the number of mixture components.

 * L 229: “ Third, the KL-divergence in our objective function acts as a regularization term, the weight of which can be tuned to reduce over-fitting” - this is also true in principle for MAP estimates (which implies computing a posterior) as in [4], the posterior can be regularized via the prior.

*Other claims*

 * L41: “they use point estimation to infer parameters”  - be more precise, which parameters?

 * Line 87 to 91: “propose to cluster task-specific parameter distributions and allow the meta-learner to select over a mixture of these distributions. Nonetheless, the meta-parameters still follow a single stationary distribution and the latent variables are inferred by point estimation, which is found prone to suffer from the catastrophic forgetting issue” - what do you mean with “follow a single stationary distribution”? As far as I can tell this refers to the mixture governed by the Dirichlet process, which is not stationary. If used like this, the word “single” also applies to the current paper (a single DPGMM distribution).

 * L94-95: “ We overcome these weaknesses by representing meta-parameters with a dynamically updated mixture model and inferring latent variables via structural variational inference.” Again, the first half or this sentence is what [4] also does, and it is unclear (from the arguments in the text) why “inferring latent variables via structural VI” (as opposed to MAP estimation?) helps overcome the weaknesses. Be more precise here.

 * Sec. 4 title - not sure what’s particularly Bayesian or non-Bayesian about the proposed model. Both, equation 1 and 2 approximate the Bayesian posterior over parameters \theta. 2 uses a hierarchical model to do so (but the parameters \phi could in principle be absorbed into a more complex parameter \theta). There isn’t really anything more or less Bayesian about it - it’s just in one notation (Eq. 2) the hierarchical model is made explicit, whereas in Eq. 1 it could either be a hierarchical model or a flat model (the latter is of course more commonly seen in the literature). I agree that the model proposed is able to capture some uncertainty that cannot be captured by e.g. [4]. I personally would find it more fitting to call the model DPGMM, Dirichlet prior **Gaussian** mixture model.

 * Line 139-140: “Consequently it will be inefficient if the task-specific parameters \phi_t are adapted from the same distribution of meta-parameters \theta_t.” This is strictly speaking not true (unless the notation is interpreted in a particular way) - the distribution of meta-parameters does not necessarily mean that there is only a single mode for \theta_t - the notation parameterizing the meta-distribution with \theta_t includes the case where \theta_t is a Gaussian mixture model or some other potentially multimodal distribution.

 * Line 144: “As meta-parameters follow Gaussian distributions” - why is this the case? Is this the modeling assumption? In general the meta-distribution should be multi-modal (which is exactly what is achieved with the mixture model), a Gaussian is clearly a wrong / overly simplifying modeling choice.

 * Line 228: “For most dissimilar tasks, point estimates can be dramatically changed, which is prone to forgetting previous knowledge”. This is to some degree also true for unimodal (Gaussian) distributions used in the paper (the only way they can cope with dissimilar tasks is by increasing their variance dramatically). The main mechanism to overcome this here (and in [4]) is via the mixture model.


**Improvements:**

1) Be more precise and very clear about differences and claimed novel advantages (see criticism above).

2) The question is where exactly the observed performance differences, particularly w.r.t. DPMM [4] come from. The following ablations are needed to determine this (the emphasis on comparing against [4] is because it is a very similar model with a different inference procedure)

2a) The proposed model reduces to DPMM if the variance of the Gaussians is manually fixed to very low values. Keeping everything else fixed, does the model perform better/worse or the same compared to DPMM? (this highlights the performance difference due to the different VI scheme).

2b) Another source of the improvements could be the annealing of one of the regularizer terms. Show a control experiment without annealing, and ideally attempt a similar annealing for the competitor methods (I am aware the latter can be tricky). It’s important to know how much of the performance gain is due to annealing.

2c) It is currently unclear how hyper-parameters for the method were found (including the parameter for the priors which act as regularizers). The performance gains might come from the fact more time went (perhaps accidentally while developing the method) into hyper-parameter tuning. Ideally show a clear process for hyper-parameter tuning and apply the same process to the competitor methods to make results more comparable.

3) Ablation: show how sensitive the results are to the various hyper-parameters. Do they need to be selected very carefully, or is the VI scheme quite robust

4) What was the (effective) number of clusters used by DPMM - is it comparable to DPGMM, and was there any attempt to tune the concentration hyper-parameter of DPMM?

5) Just to be sure (and since the writing might suggest otherwise) - the DPMM implementation that is being compared against is using the Dirichlet process (as in Section 5 of [4]) and not the static model (as in the first part of [4])? How was the DPMM implementation obtained?

7) Meta-comment: the benefits of the method (compared to point-wise meta-parameters as in [4]) should really become apparent when multiple different, but related tasks belong to the same cluster. This might or might not happen implicitly in the experiments shown (implicit subtasks within each dataset might exist). It would be interesting to see this clearly in a controlled experiment with synthetic data.






**Comments:**

 * Line 206: “Variational inference of \phi” - a point-wise global initialization is used in conjunction with (a few steps of) gradient descent. This sounds very similar to what [4] does via MAML. So what’s the difference between this and [4]? The appendix points out this similarity to MAML, what’s important to distinguish from [4] is whether the sample is drawn from the Gaussian or simply its mode/mean is selected for the global initialization (which would correspond exactly to [4]). The pseudocode in the appendix confirms that the value is sampled. Since this is one of the main differences this needs to be pointed out more clearly in the main text.
 * Compare (discuss, not necessarily via experiments) against “Hierarchical Indian Buffet Neural Networks for Bayesian Continual Learning” (Kessler, UAI 2021). What are advantages and disadvantages of the construction proposed in this paper? (I am not an author btw.)
 * Define in the intro: what is a ‘low resource’ task?
 * Error-bars in Table 2 often overlap between the first and second best solution. Please point this out in the text.
 * Another pass to check grammar and spelling
 * Line 185: “ Specifically, we use handy Dt to recursively infer latent variables” I don’t understand this sentence.

**Time Spent Reviewing:**

4.5

---

> ### Author Response · Authors · 2021-08-09
> **Response to the concerns of claimed advantages and ablation studies**
>
> ----Overall response:
>
> We thank the reviewer for the detailed comments and constructive suggestions and believe all the issues mentioned can be properly addressed in the final version. The major concerns lie in two aspects: (i) the properly claimed differences between our model and the reference work [4], and (ii) ablation and control experiments to back up some claims. We take this chance to clarify these issues.
>
> In terms of the differences between our model and the work [4], we rephrase the Discussion at lines 219-230 to correct potentially misleading claims. We analyze three major differences that consequently lead to performance improvement. First, the meta-parameters in our model are represented by a mixture of Gaussians compared to delta distributions in [4]. Compared to delta-distributions, Gaussians generally enable a larger capacity in tackling related but moderately different tasks in one cluster, and can capture some uncertainty that delta-distributions with zero variance cannot do. Second, as we said in the paper, “we derive a structural variational inference method to approximate the posterior while [4] only provides point estimates based on maximum a  posteriori (MAP). In a streaming setting, point estimates on the past cannot imply doing well on future data as underlying generation functions can be ambiguous even given some prior information.” Third, despite in principle the posterior can be regularized by the prior in [4], the reference work [4] neither explicitly explains how to achieve this goal nor adopts this regularization method in their model (see Algorithm 1 in [4]). In contrast, the KL-divergence in the VI framework in our model can be naturally considered as the regularization term. By tuning the weight of this regularization term, we can reduce overfitting to the incoming data, leading to improved performance (and we have proved this in the Experiments section).
>
> As for ablation and control experiments, we followed the suggestions and provided detailed comparison results in the following Ablation Study in Improvement.
>
> ----Criticism w.r.t. claims
>  1. Thanks for pointing out the misleading claims at Line 221. We will correct them with the above explanatory sentences.
>
>  2. We would like to emphasize that despite in principle the posterior can also be regularized by prior in [4], this regularization method is not mentioned or adopted in their model (see Algorithm 1 of [4]). So whether or not using the regularization term is indeed an important difference between our model and the model in [4], which also leads to the difference of their performance.
>
> 3. At Line 41, the reference work [4] used point estimation to infer model parameters, including meta-parameters and task-specific parameters.
>
> 4. At Lines 87 to 91, we will rephrase this sentence and emphasize the meta-parameters in [4] follow delta distributions and are actually obtained by point estimation.
>
> 5. At Lines 94-95: We analyze the weakness of point estimates at Lines 224-227, “In a streaming setting, point estimates on the past cannot imply doing well on future data as underlying generation functions can be ambiguous even given some prior information.” As opposed to MAP, the structured VI can efficiently approximate the posterior of meta-parameters and is more robust in dealing with task ambiguity and heterogeneity.
>
> 6. Section 4 Title: For a general continual learning problem without the limitation of low-resource data, model parameters can be just denoted by $\theta$, as indicated in Equation-1, and it is not necessary to distinguish meta and task-specific parameters. While with the limitation of low-resource data, we rely on meta-learning that focuses on such distinctions in Equation-2. The term “Bayesian” is used to describe the approach to meta-learning, where meta and task-specific parameters have their own distributions and are learned in a Bayesian way, as opposed to non-Bayesian meta-learning.
>
> 7. At Lines 139-140: Most commonly, researchers assume simple, often unimodal Gaussians as the variational distribution family for meta-parameters. By “the same distribution”, we actually want to underline the simple Gaussian distributions in prior works, as opposed to a mixture of Gaussians in our work. We will clarify this sentence in the final version.
>
> 8. At Line 144: We will remove this phrase to avoid confusion.
>
> 9. At Line 228: Compared to point estimates, Gaussian samples are able to capture uncertainty, and can be more robust in dealing with dissimilar tasks, as is mentioned in the review, by increasing distribution variances. We agree the main mechanism to overcome forgetting is via the mixture model and will rephrase this sentence to make it clearer.
>
> ----Improvements
> 1. See the overall response.
>
> 2. Ablation study
> We followed the suggestions of the reviewer and conducted ablation experiments to analyze the reasons for the performance difference between our model and DPMM. There are three variants of our VC-BML:  VC-BML-Delta, VC-BML-MAP, and VC-BML-NOR. The experimental results are shown in the following table.
> | Model        | Omniglot     | CIFAR-FS     | miniImagenet  | VGG-Flowers  |
> |--------------|--------------|--------------|---------------|--------------|
> | DPMM         | 99.33 ± 0.28 | 83.52 ± 1.21 | 73.96  ± 1.37 | 70.09 ± 1.40 |
> | VC-BML-Delta | 99.48 ± 0.16 | 84.43 ± 1.05 | 75.45 ± 1.33  | 71.27 ± 1.52 |
> | VC-BML-MAP   | 99.50 ± 0.16 | 84.98 ±1.11  | 77.46 ± 1.38  | 76.27 ±1.51  |
> | VC-BML-NOR   | 94.81 ±0.51  | 80.76 ± 0.99 | 70.38 ± 1.46  | 66.88 ± 1.32 |
> | VC-BML       | 98.62 ± 0.38 | 86.95 ± 1.15 | 78.29 ± 1.34  | 79.37 ± 1.43 |
>
>     2.1 We first examine the contribution of GMM over a mixture of delta distributions by comparing VC-BML-Delta and VC-BML. VC-BML-Delta fixes the variance of Gaussian distributions to be a low value (10^{-5}) and converts GMM to a mixture of delta distributions. We find that the performance of DPMM and VC-BML-Delta is quite similar, but lower than that of VC-BML on three datasets. This comparison quantifies the contribution of GMM over a mixture of delta distributions.
>
>      2.2 We then examine the advantage of VI over MAP. We developed VC-BML-MAP that replaces the structured VI with MAP. From the table, we find that VC-BML outperforms VC-BML-MAP on three out of the four datasets. The performance difference shows the quantified contribution of VI in our model.
>
>     Besides, our model is worse than VC-BML-Delta and VC-BML-MAP on the Omniglot dataset. This is due to the influence of KL-divergence and its weight in the VC-BML. Specifically, VC-BML-Delta does not suffer from the influence of the KL-divergence because the  KL-divergence is near zero when the Gaussian distribution collapses to a delta. As for VC-BML-MAP, there is no KL-divergence in MAP. These comparisons motivate us to properly weigh the KL-divergence in the VC-BML model, which consequently produced better performance on Omniglot as we showed in Figure 5 in the main text. Besides, in Table 2 of the Appendix, we show the optimal weights of KL-divergence that leads to the best performance on all datasets.
>
>     2.3 To further examine the effects of weighting KL-divergence, we introduced VC-BML-NOR that does not tune the weight of KL-divergence and set it to be 1. We notice that VC-BML-NOR performs much worse than VC-BML. So if the KL-divergence is not properly scaled, the model is actually learning nothing but forcing the posterior to collapse to the prior, leading to the severe mode collapse problem. We discuss this over-regularization effect of the KL-divergence in Section 5.3 of the main text.
>
>     2.4. Hyper-parameters: We used grid search to tune the hyper-parameters of our model and baselines based on the validation dataset. Please see Section E.1 and E.3 of Appendix for more details. The performance improvements are therefore not due to more efforts paid to tune the hyper-parameters of our model.
>
> 3. Number of clusters in DPMM: After training DPMM on the four experimental datasets, there are actually 4 task clusters, which is the same as the effective number of clusters in our model (although we set the upper bound of cluster number to be 6), as shown in Figure 4. We tune the concentration hyperparameter of DPMM in a similar way as our model. Please see Section E.1 of Appendix for more details of the experimental set-ups.
>
> 4. Implementation of DMPP: We used the Dirichlet process Mixture Process with dynamically increasing mixtures instead of the static model. Since the authors of DPMM do not publicly release their code, we implemented DPMM by ourselves based on the DPMM paper.
>
> ----Comments
> 1. Similar to our work, the paper “Hierarchical Indian Buffet Neural Networks for Bayesian Continual Learning” focuses on the Bayesian approach to continual learning and developed a structured variational inference scheme. The Dirichlet Process is used to adaptively determine the required resources (neurons of each layer in that paper while Gaussian distributions in our work). In that paper, each neural layer can have multiple resource units (i.e., neurons) and it uses Indian Buffet Process (IBP) to automatically determine the number of neurons in each layer. In contrast, we use the Chinese Restaurant Process that indicates one task belongs to a task cluster and can be tackled by one resource unit (i.e., a mixture component of the meta-parameter distributions). It is interesting to try IBP to break the one-to-one correspondence between the task clusters and mixture distributions: a task can be tackled by a subset of mixture components. We leave it in future works.
>
> 2. We define low-resource tasks in the first line of Introduction: tasks with scarce labeled data such as few-shot image classification.
>
> 3. At Line 185: At each timestep $t$, we use only the dataset $D_t$ at hand to infer latent variables including $\theta_t$ or $\phi_t$. We do not revisit previous datasets $D_{1:t-1}$ for such inference.

---

> > ### Comment · Reviewer_h2Cd · 2021-08-14
> > **Impressed and happy to see the thorough ablations, and responses to concrete criticism**
> >
> > I want to thoroughly thank the authors for the work and effort put into their response. I am more than happy to see the ablations - they crucially point out the contribution of each technical improvement to the improvement in scores. The ablation results are interesting and confirm that all three innovations play a role and lead to individual improvements that additively combine in the full method.
> >
> > I am also very happy that the authors responded in detail to my 'criticism' of individual lines/claims in the text. The author's responses are what I was expecting/hoping for and I want to strongly encourage the authors to incorporate this feedback into the camera-ready version. Be as precise and technical as possible, especially w.r.t. similarities/differences with previous work. I think the paper can now clearly point to particular technical improvements (which have a clear theoretical motivation) and quantitatively show how these improvements lead to better results. This will make it much easier for readers (less guesswork, vague claims) and also for adoption/follow-up work (clear 'credit-assignment' to different improvements).
> >
> > All main issues have been addressed - having a formal/mathematical definition of a low-resource task would be nice, but not crucial for acceptance.
> >
> > Overall, I am happy to raise my score and argue in favor of accepting the paper. I want to thank the authors again for putting in the work and effort, and want to encourage them to keep up this attitude towards the camera-ready version (maybe even get the synthetic-data experiment suggested in my improvement 6 in).

---

### Official Review · Reviewer_U7b8 · 2021-07-17

**Rating:** 7
**Confidence:** 3

**Summary:**

This paper attempts to solve the problem of meta-learning when data arrive sequentially. Unlike previous approaches, the authors propose a method to maintain a dynamic Gaussian mixture model over the meta-parameters. The number of clusters is picked using a Chinese restaurant process. Various experiments have been conducted to show the accuracy and superiority of the model.

**Limitations And Societal Impact:**

Adequate

**Main Review:**

This paper attempts to solve the problem of meta-learning when data arrive sequentially. Unlike previous approaches, the authors propose a method to maintain a dynamic Gaussian mixture model over the meta-parameters. The number of clusters is picked using a Chinese restaurant process. Various experiments have been conducted to show the accuracy and superiority of the model.

It is great to see a truly Bayesian treatment for meta-learning. Maintaining a multimodal latent distribution for meta-parameters is a neat idea. I did not check all the derivations but the paper follows the standard structured variational inference derivations and looks legit. Though using Chinese restaurant process for selecting the clusters is ideal, I am wondering if it is overkill. Possibly, we could have made the number of clusters a parameter of the mixture and learn it through variational inference itself. In that case, the mixing coefficients could follow a Dirichlet distribution the parameters of each Gaussian would be Gaussian-Wishart.

I would like the authors to discuss more about the practical aspects of the algorithm. What is the dimensionality of latent spaces?, how do we avoid mode collapse?, what is the computational time?, etc. If our initial distribution is not broad and diverse enough, I am skeptical if the posterior distribution can indeed pick the necessary modes of the new distribution of incoming data. As another limitation, if the parameters of the new underlying dataset are truly multimodal [1], the variational approximation would not be able to capture the variational distribution as it would pick only one mode by construction.

Minor:
* Line 58 - “It’s” - avoid contractions
* Line 101 - Use parentheses as {(x,y)} to show that what repeats is the input-output pair.

[1] Itkina et al., "Evidential Sparsification of Multimodal Latent Spaces in Conditional Variational Autoencoders," NeurIPS'20


**Time Spent Reviewing:**

1

---

> ### Author Response · Authors · 2021-08-09
> **Clarifications of the practical concerns of the algorithm**
>
> We thank the reviewer for the positive comments and the insightful questions, as well as the helpful suggestions. The following are our responses to the questions mentioned in the comments.
>
> 1. Making the number of clusters a parameter of the mixture is an interesting idea. One concern is that learning this parameter through variational inference (VI) increases the complexity of the VI scheme, thus we are not sure if VI is sensitive to new data and can robustly obtain proper parameter values. Besides, due to the unpredictable task complexity in the online dynamic setting, we believe it is appropriate to use the Chinese Restaurant Process in order to adaptively determine the number of clusters.
>
> 2. As the latent variables in this paper are meta-parameters and task-specific parameters, the dimensionality of the latent space is actually determined by the number of parameters in the deep neural network. In particular, we define a CNN architecture and present its details in Table 1 in the Appendix. Roughly, there are 112,000  parameters in the defined CNN. So the dimensionality of the latent space is about 112,000.
>
> 3. We report the computational time of our model at Lines 174-176 in the Appendix.
>
> 4. Mode collapse:  If the initial distribution is not adequately diverse, our model may not be able to capture all data modes. In order to reduce the strong loss caused by the data of collapsed modes, there are two methods. First, our model will adaptively add new components to the mixture distributions, and the new components will focus on collapsed modes to reduce relevant losses. Second, we can adjust the ELBO function to avoid the posterior collapsing to the prior. Specifically, we can use the hyperparameter $\nu$ to down-weight the KL-divergence term in ELBO. Smaller $\nu$ encourages our model to find the mode of the true posterior of the incoming data instead of collapsing to the prior.
>
> It is true that if a dataset is heterogeneous and requires multiple modes in meta-parameter distributions, the developed VI can only pick one mode. To address this limitation, we can replace the Chinese Restaurant Process with the Indian Buffet Process, which uses multiple component distributions to tackle a single task. We leave this for future work.

---

### Official Review · Reviewer_SRM6 · 2021-07-26

**Rating:** 6
**Confidence:** 3

**Summary:**

In this paper, the authors propose Variational Continual Bayesian Meta-Learning (VC-BML) to handle various tasks which arrives sequentially. In particular, the authors consider the tasks follow a non-stationary distribution and assume that the previous dataset cannot be stored in memory. To deal with the non-stationarity and low-resource, the authors propose to perform sequential Bayesian inference where the meta-parameters are estimated by Dynamic Gaussian Mixture Model (DGMM) whose number of components are determined by Chinese Restaurant Process.
In the experiments, the authors demonstrate VC-BML successfully learns the new task without suffering the negative knowledge transfer issue while avoiding the catastrophic forgetting and maintaining the ability to solve the previous tasks.

**Ethical Concerns:**

No issue

**Limitations And Societal Impact:**

I agree with the limitations raised by authors, and feel they are adequate.

**Main Review:**

The paper is well organized and the idea is easy to follow.
The method is technically sound; a meta-learning framework is combined with DGMM and Chinese Restaurant Process which are well established method to deal with the unknown number of components.
The drawback of the method is that the method has to keep the meta-parameter for each task, which becomes difficult when the number of tasks increase.
The experimental results are convincing as the proposed method is compared with the baselines and existing methods including DPMM and BOMVI. It shows the benefit to combine the Chinese Restaurant Process.

I have a few minor comments.
1. D_t^S appears in Figure 1 and Equation (15) without any explicit explanation. Please add explanation on D_t^S.
2. EBLO should be ELBO at Lines 205, 261, and 311.

**Time Spent Reviewing:**

6hours

---

> ### Author Response · Authors · 2021-08-09
> **Explanation of the support set and the query set**
>
> We thank the reviewer for the detailed comments and helpful suggestions. The following is our responses to the comments.
>
> 1. $\mathcal{D}_t$ is split to a support set $\mathcal{D}_t^S$ for adapting task-specific parameters $\phi_t$ and a query set $\mathcal{D}_t^Q$ for task evaluation. Specifically, we use amortized variational inference to compute variational parameters of $\phi_t$ from $\mathcal{D}_t^S$. The details of this procedure is provided at the end of Section C in the Appendix. Also, we will add an explanation in the main text.
>
> 2. We will correct the typos.

---

### Official Review · Reviewer_THpm · 2021-08-01

**Rating:** 7
**Confidence:** 3

**Summary:**

The authors tackle the problem of continual meta-learning, where the tasks are streamed. This settings comprises the additional challenge of catastrophic forgetting.
They introduce a training method VC-BML, which updates a GMM over the meta-parameters dynamically. The idea is to cluster the posterior meta parameters over the modes of the GMM. Additionally, they leverage structured variational inference in order to approximate the posteriors.
The experiments show an improvement in terms of mean meta test accuracy over the stream of tasks.

**Limitations And Societal Impact:**

The authors have adequately addressed the limitations and potential negative societal impact of their work.

**Main Review:**

# Originality
1- **Are the tasks or methods new?**

The method is new overall.

2- **Is the work a novel combination of well-known techniques?**

The work builds up on techniques from variational inference and meta learning, overall it's not a simple combination of well-known techniques.

3- **Is it clear how this work differs from previous contributions?**

The difference from previous contributions is discussed thoroughly.

4- **Is related work adequately cited ?**

The related works are adequately cited, to my knowledge.

# Quality
1- **Is the submission technically sound ?**

The submission is technically correct.

2- **Are claims well supported (e.g., by theoretical analysis or experimental results) ?**

- The theoretical analysis is correct and supports the paper's motivation.
- The experiments are clearly split and support the claims of the paper.

3- **Are the methods used appropriate ?**

- The methods used are tailored to the problem (DGMM, variational inference ....)

4- **Is this a complete piece of work or work in progress ?**

It is a complete piece of work.

5-  **Are the authors careful and honest about evaluating both the strengths and weaknesses of their work ?**

The authors also evaluate the limits of their work in terms of :
- The importance of the KL down-weighting parameter
- How to select the maximum number of mixtures

# Clarity:

1- **Is the submission clearly written?**

The submission is easy to follow and clearly written. Also, the theoretical analysis is clearly outlined.

2- **Is it well organized? (If not, please make constructive suggestions for improving its clarity.)**

The paper is well organized and easy to follow.

3- **Does it adequately inform the reader? (Note that a superbly written paper provides enough information for an expert reader to reproduce its results.)**

- The source code is provided.
- The hyperparameters are also presented in the appendix.

# Significance:
1- **Are the results important?**

Beyond the improvement of the state of the art, the results are important in the way they validate the DGMM for meta learning as a reliable proof of concept.

2- **Are others (researchers or practitioners) likely to use the ideas or build on them?**

It is likely that practitioners may apply the method to real world systems, given the improvement in comparison with previous methods.

3- **Does the submission address a difficult task in a better way than previous work?**

The results in Table 2 show that the proposed method outperforms previous works on most settings presented.

4- **Does it advance the state of the art in a demonstrable way?**

It advances the state of the art, though the improvement is around 1 to 2 percents in terms of meta-test accuracy.

# Minor comments
- There is a typo in the page 2 of the Appendix, line 29 "is infer" -> "is to infer" :)

**Time Spent Reviewing:**

15

---

> ### Author Response · Authors · 2021-08-09
> **Remove typos**
>
> We thank the reviewer for the detailed review and the positive comments. We will remove the typo and make the paper clearer.

---

### Decision · Program_Chairs · 2021-09-27

**Decision:**

Accept (Poster)

**Comment:**

The paper proposes a new, fully-Bayesian method for online meta-learning: VC-BML. A mixture models over the meta-parameters is updated dynamically. This is used as an informative prior for the upcoming tasks.

The Reviewers highlighted the novelty of the method with respect to previous works. The paper is theoretically sound, and VC-BML improves over known meta-learning algorithms in practice. Therefore, I will recommend to accept the paper.

Please take into account all the comments of the Reviewers for the camera-ready version of the paper. Especially, Reviewer h2Cd pointed out a few problems in the writing of the paper and in the comparison with Jerfel et al. 2019. The Reviewer updated his/her evaluation of the paper following your detailed reply and your promise to revise the paper accordingly.